# The effects of task similarity during representation learning in brains and neural networks

Nicholas Menghi [1] ✉, W. Jeffrey Johnston [2], Simone Vigano' [1,3], Max Andreas Bosse Hinrichs[1], Burkhard Maess [1], Stefano Fusi [2,4] & Christian F. Doeller [1,5]

The complexity of our environment poses significant challenges for adaptive behavior. Recognizing shared structures across tasks can theoretically improve learning through generalization. However, how such shared representations emerge and influence performance remains poorly understood. Contrary to expectations, our findings revealed that individuals trained on tasks with similar low-dimensional structures performed worse than those trained on dissimilar tasks. Magnetoencephalography revealed correlated neural representations in the same-structure group and anticorrelated ones in the different-structure group. Crucially, practice reduced this performance gap and shifted the neural representations of the tasks in the same-structure group towards anticorrelation, resembling those in the different-structure group. A neural network model trained on similar tasks replicated these findings: tasks with similar structures require more iterations to orthogonalize their representations. These results highlight a complex interplay between task similarity, neural dynamics, and behavior, challenging traditional assumptions about learning and generalization.

To efficiently interact with the environment, intelligent agents create and use internal models of the world that represent the association between sensory inputs and selected actions or decisions[1,2]. These representations might be built by simply selecting a single task-relevant feature (and thus suppressing or ignoring irrelevant ones) or by implementing more complex operations to extract the abstract structure of the task at hand[3–6]. The task structure defines a set of parameters that organize one or more tasks. It creates a compressed space with lower dimensionality than the original sensory input while preserving a similar amount of information[7]. Utilizing this structure enables a more efficient mapping between inputs and actions[8,9]. We can, for example, learn that there is a relationship between the amount of light and water and the growth of the plants in our garden without

having to memorize all the possible combinations of features and related outcomes. Given the extraordinary load on the cognitive system created by the multitude of tasks we learn, a particularly effective strategy is to capitalize on the shared similarities between different task representations, thus helping the brain to efficiently learn and generalize through a wide range of situations and experiences. This identification of commonalities operates at varying levels of abstraction. For example, there is evidence of shared representations for number and space[10], imagery and visual processing[11], face and object perception[12], sequences in working memory[13], associative learning[14,15] and for the representations of self and others in social cognition[16]. Such mechanisms have been observed in machine learning as well. Neural networks trained to learn multiple tasks also exploit similarities

[1]Department of Psychology, Max Planck Institute for Human Cognitive and Brain Sciences, Leipzig, Germany. [2]Center for Theoretical Neuroscience and Mortimer B. Zuckerman Mind, Brain, and Behavior Institute, Columbia University, New York, NY, USA. [3]Center for Mind/Brain Sciences, University of Trento, Rovereto, Italy. [4]Kavli Institute for Brain Sciences, Columbia University, New York, NY, USA. [5]Kavli Institute for Systems Neuroscience, NTNU, Trondheim, Norway. ✉e-mail: menghi@cbs.mpg.de

by creating a low-dimensional representation common to different tasks benefiting learning and generalization[17–21]. Going back to our garden, we can infer that plants that appear similar to known plants might also have similar water and light requirements.

Recently, the investigation of how task representations are acquired and influence generalization, usually referred to as "representation" and "transfer" learning, respectively, has gained momentum in cognitive neuroscience and machine learning[14,19,21–24]. Few studies have started to elucidate the underlying brain mechanisms in humans, for instance observing neural (EEG) signatures of the emergence of low-dimensional, task-relevant representations[25–27], or how their alignment can facilitate transfer learning across domains that are based on magnitude or linearly ordered structures[28].

However, a clear understanding of how the human brain creates and uses these representations remains elusive and a central topic of current research[29,30]: To fill this gap, we designed an experiment where we investigated the formation and use of shared representations across different tasks during learning, while at the same time monitoring the neural activity of participants using magnetoencephalography (MEG). Participants learned two tasks (named "Conceptual" and "Spatial", see "Methods") with an interleaved training regime. In both tasks, they had to decide whether a fictitious plant seed would grow or die based on either the relative amount of water and light ("Seed1", Conceptual task, Fig. 1A top, where the opacity of the corresponding symbol indicates the amount of a feature [i.e., water or sun]) or the X-Y spatial position in visual space on a computer screen ("Seed2", Spatial task, Fig. 1A bottom). To efficiently perform this classification, participants had to discover the underlying low-dimensional, task-relevant, hidden structure capturing the correct ratios of the two conceptual (water and light) or spatial (x and y) features (Fig. 1B). Crucially, participants were divided into two groups. In one group, the underlying low-dimensional, task-relevant hidden structure of the two tasks was the same ("Same Structure" or

SameSt group, Fig. 1B top row), while in the other group, they were orthogonal ("Different Structure" or DiffSt group, Fig. 1B bottom row)(see "Methods"). Both groups performed a training phase first, in which participants received trial-wise feedback in a classification task, followed by a test phase, where they were presented with both old and new stimuli to classify without feedback to test their ability to generalize.

We expected the SameSt group to show higher performance than the DiffSt group in learning to perform the two tasks (training phase), as they share the same underlying hidden structure. We predicted that this advantage should also lead to better generalization performance in the test phase. In parallel, we analyzed the MEG signal to gain insight into the emergence and use of task-relevant representations and their differences across the two groups.

Contrary to our expectations, the SameSt group performed worse than the DiffSt one during the training phase, indicating that the shared structure between tasks led to interference rather than facilitation. MEG analyses revealed that the representations of the two tasks were initially correlated in the SameSt group but anticorrelated in the DiffSt group. With practice, the gap in performance decreased, and the neural representations of the two tasks in the SameSt group shifted toward an anticorrelated pattern similar to that of the DiffSt group. This dynamic aligns with findings from the cognitive control literature, suggesting mechanisms by which the brain adapts to interference through representational reorganization[31–33]. In the Discussion, we will further compare these findings with studies on cognitive control, highlighting both similarities and key differences in how interference emerges and is resolved across different paradigms.

Finally, we trained neural networks to perform tasks analogous to the experimental tasks. Comparing the performance and learning dynamics of the trained networks with those of human participants allowed us to draw conclusions about the cognitive mechanisms involved.

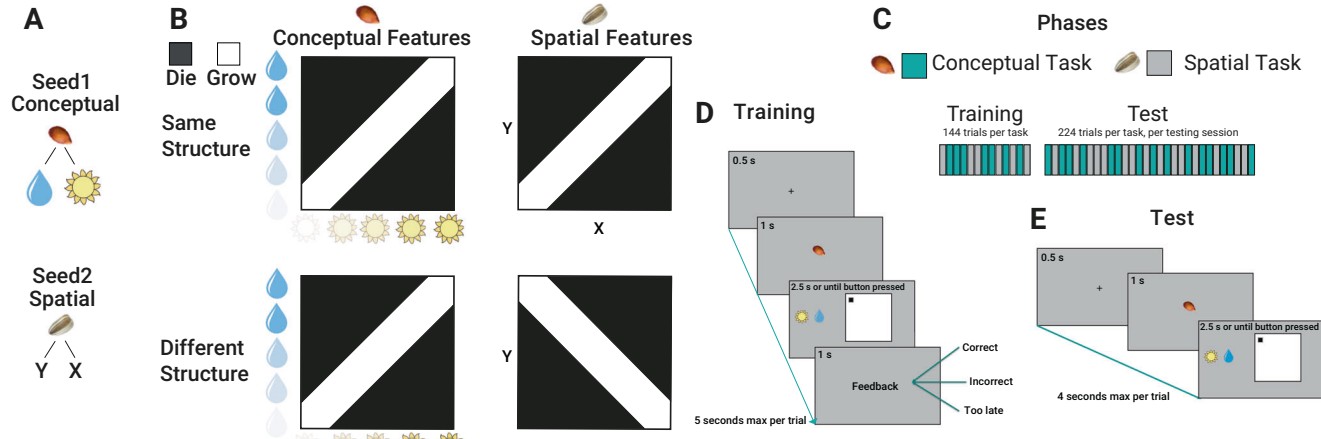

Fig. 1 | Stimuli, Task Structures and Trial Structure. A Shows the experimental stimuli and cues. Two seeds were used to cue for the two different tasks, spatial and conceptual. In the conceptual task, we used a sun and a water drop symbol as features composing a stimulus configuration. In the spatial task, we used a position (x,y) on a square map as a stimulus. B shows the different structures used. In the conceptual task, each vertical (sun opacity) and horizontal (a drop of water opacity) position can be combined, creating a continuous map or structure from which we draw stimuli. In these structures, black color is associated with the outcome "die", and white color is associated with the outcome "grow". The structures of the two tasks were the same in one group and orthogonal in the other group of participants. C shows the different phases of the experiment. In each phase, conceptual and spatial tasks were organized in an interleaved regime. D shows the schematic of a trial structure during the training phase. A fixation

cross was shown for 0.5 s, and then the seed cueing the task domain appeared for 1 s. Brown seed cued participants to pay attention to the conceptual features and ignore the spatial ones, and vice-versa, the gray seed cued participants to pay attention to the spatial features and ignore the conceptual ones. Afterward, both spatial and conceptual configurations appeared on screen and stayed there for 2.5 s maximum, or until response. This is when participants used a button box to predict if, based on the relevant features presented, the seed was associated with "grow" or "die". Finally, feedback appeared and stayed on screen for 1 s. E shows the schematic of a trial structure during the testing phase. A fixation cross was shown for 0.5 s, and then the seed cueing the task domain appeared for 1 s. Afterward, spatial and conceptual configurations appeared on screen and stayed there for 2.5 s maximum, or until response. Differently from the training phase, no feedback was provided.

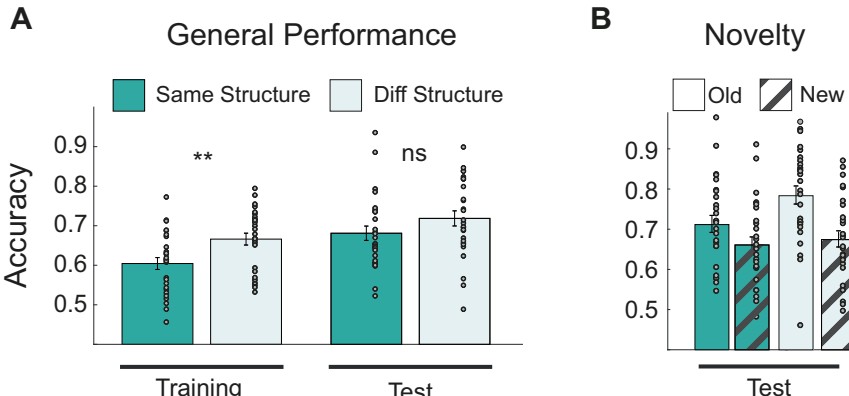

**Fig. 2 | General performance and generalization. A** shows participants' accuracies during the training and test phases. Participants are divided into SameSt ($N = 27$) and DiffSt ($N = 27$) group conditions. Statistical significance was assessed using a two-sided paired t-test. SameSt (mean = 0.6044, SD = 0.0783) and DiffSt (mean = 0.6661, SD = 0.0770) differed significantly, t(52) = −2.91, $p = 0.005$, Cohen's d = 0.80, 95% CI [−0.104, −0.019]. Asterisks indicate significance levels (*$p$ < 0.05, **$p < 0.01$, ***$p < 0.001$) (**B**) shows participants' performance for old and new stimuli during the testing phase. Participants are divided into SameSt and DiffSt following the color scheme of (**A**). Data are presented as mean values, error bars represent standard error of the mean. Source data are provided as a Source Data file.

## Results

We trained participants to perform an associative learning task across two contexts in an interleaved regime. They had to predict if a seed was going to "grow" or "die" based on two distinct sets of features, where the relevant features were determined by the identity of the seed (see Fig. 1A, B). Decisions about the first seed relied on what we called the "conceptual features": participants had to learn the association between configurations of the opacity of a sun and a drop of water images and the outcome (grow or die). Decisions about the second seed relied on "spatial features": participants had to learn the association between configurations of position in space in which the seed was planted and an outcome (grow or die). Two different seeds cued the different contexts (See Fig. 1A. Participants were randomly assigned to either a "Same structure" (SameSt) or a "Different structure" (DiffSt) group, which differed based on the similarity between the feature to outcome mappings (see Fig. 1B). Participants in the SameSt group had the same feature-outcome structure in both contexts, while participants in the DiffSt group had a flipped structure across the two contexts (Fig. 1B). Each participant was first trained via feedback learning (see Fig. 1D), and then tested (see Fig. 1E) during two separate sessions, within the same day, separated by a navigation task, that pertained to a different question and will be analysed separately (See "Methods").

To assess the emergence of representation of and across tasks, we divided the analysis into two parts. First, in our behavioural data analysis, we compared participant accuracies between the two distinct groups and assessed their generalization performance on a novel set of configurations that were interleaved with old ones. Second, for our analysis of the MEG data, we employed Representation Similarity Analysis (RSA) to establish links between the neural representations and participants' behavioural results.

### Behavioural results

We started by analyzing behavioral performance during the training phase, where accuracy was computed as the proportion of correct responses over all trials across the two tasks (Spatial and Conceptual). A 2 × 2 mixed-design analysis of variance (ANOVA) revealed a main effect of group (DiffSt vs SameSt; F(1,52) = 8.455, $p = 0.005$), but no main effect of task (Conceptual vs Spatial; F(1,1) = 0.145, $p = 0.704$) nor interaction (F(1,52) = 0.246, $p = 0.621$). Post-hoc tests revealed that, contrary to our expectations, participants belonging to the DiffSt group performed better compared to the one in the SameSt group

(t(52) = 2.908, $p = 0.005$; t-test)(Fig. 2A, left). This difference, however, was reduced and no longer statistically significant during the test phase (t(52) = 1.426, $p = 0.159$)(Fig. 2A, right). To further examine whether participants learned the decision boundary, we conducted an additional analysis testing whether accuracy varied as a function of stimulus distance from the decision boundary. The results indicate that performance improved with increasing distance, suggesting that participants were sensitive to the underlying task structure. This pattern remained stable across both training and test phases, reinforcing the idea that participants learned a structured representation of the task. Full details of this analysis can be found in the Supplementary Fig. 1.

A detailed analysis of performance during the test session through a three-way (2 × 2 × 2 structure group by domain by novelty) mixed-design ANOVA revealed no main effect of group (DiffSt vs SameSt; F(1,52) = 2.494, $p = 0.120$) or domain (Conceptual vs Spatial; F(1,52) = 1.250, $p = 0.268$), but a strong main effect of novelty (New vs Old configurations; F(1,52) = 37.14, $p < 0.001$), indicating that participants were more accurate when classifying previously encountered stimuli (confirmed with a post-hoc t-test, t(26) = 5.864, $p < 0.001$). Performance on new stimuli was similar across groups, showing that both SameSt and DiffSt participants generalized knowledge to novel configurations. The only significant interaction was between Group and Novelty (F(1,52) = 5.237, $p = 0.026$), showing that the DiffSt group classified old stimuli more accurately than the SameSt group (approaching significance in a post-hoc t-test, t(52) = 2.040, $p = 0.051$). Despite this difference in remembering previously seen stimuli, both groups successfully classified new stimuli above chance level (mean classification accuracy: 0.66; t(53) = 11.898, $p < 0.001$), with no significant difference in generalization ability between them (t(26) = −0.376, $p = 0.709$). These results are summarized in Fig. 2B.

In short, contrary to our expectations, we found that similar structures between tasks did not facilitate learning, and this was reflected in lower classification accuracy. This gap seemed to be reduced with practice, as both groups reached equivalent classification and generalization performance during the test phase. To gain a better understanding of this observation, we analysed the corresponding MEG data.

### MEG results

While participants were performing the experiment, we measured their brain activity non-invasively using MEG. To gain insight into

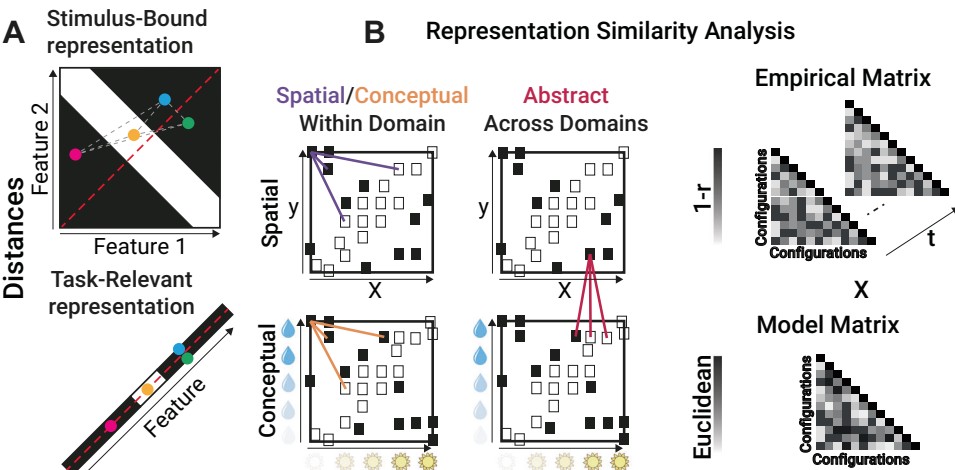

**Fig. 3 | Model distances. A** illustrates an example of distance computations for four points, according to Stimulus-Bound and Task-Relevant representations. The four points are selected here solely for visualization purposes. Stimulus-bound distances are calculated within a two-dimensional feature space, accurately capturing the configurations of conceptual and spatial features. Gray dashed lines denote distances between points. Task-relevant distances, however, are computed within a compressed space optimized for learning the task structure. These distances are measured along a one-dimensional manifold, and points are projected onto the red dashed line by subtracting feature 2 from feature 1, which retains essential task information critical for generalization. Here, the blue and green points overlap, reducing their distance to zero. **B** shows a schematic of the calculation of distances for spatial, conceptual, and cross-domain tasks (example from the SameSt group). The matrices presented in the figure are schematic examples meant to illustrate

how distances are calculated across configurations for within- and cross-domain tasks. Each white and black square in the four-square maps represents different configurations, with white indicating "grow" and black indicating "die." Within a domain, Euclidean distances were calculated for spatial or conceptual tasks. Across domains, distances between the spatial and conceptual task stimuli were computed as if both lay in the same space. The red, orange, and purple lines indicate specific example distances used in the analysis, respectively, distance across domains, within domain in the conceptual space and within domain in the spatial space. This approach yielded six model matrices in total: three stimulus-bound and three task-relevant. We then computed correlations between these model matrices and empirical matrices, where distance was calculated as the correlation distance between the Event-Related Fields (ERF) of different configurations at each time point.

how task representations emerge and support behavior, we could analyze the representational geometry of task structure by means of RSA[34]. In particular, we asked whether, how, and when representational models of how the different stimuli related to each other emerged in neural activity. By looking at such relational representations, we aimed to elucidate how task information is structured in the brain and understand what generated the observed interference in behavior. We generated six models encapsulating two distinct representations of the task: stimulus-bound and task-relevant (see Fig. 3A). The first model, stimulus-bound, captured the similarity between stimuli in the bi-dimensional stimulus space of relevant features, and was calculated by computing the 2D Euclidean distances between stimuli, either conceptual (defined by the amount of water and sun) or spatial (defined by x and y coordinates on the screen); the second model, task-relevant, reflected the similarity between stimuli in the compressed, low-dimensional space that captures the correct ratio between the two features necessary for classification, and it was computed as the 1D Euclidean distance between stimuli after they have been compressed by projecting them according to their subspace. These models were constructed for both the stimuli within the spatial and conceptual domains (Fig. 3B top and bottom left panels), as well as for their cross-domain or "abstract" correspondence (see Fig. 3B top and bottom right panels, for a description of these procedures see the Methods section). These models were correlated with neural matrices where dissimilarities were expressed as 1-Pearson's r between the activity patterns across MEG channels associated with the presentation of each stimulus at a given time point. A positive correlation between neural activity and one of these models means that the brain's response patterns correspond to the model structure. For example, a positive correlation with the stimulus-bound model would indicate that neural activity is organized according to the physical properties of the stimuli (such as their spatial or conceptual features). In contrast, a positive correlation with the task-relevant model would suggest

that neural activity is more closely aligned with the compressed representation capturing the task structure.

Statistical significance was assessed non-parametrically at the group level using a cluster-based permutation approach with a cluster-forming threshold of $p < 0.05$ (two-tailed), and a corrected significance level of $p < 0.05$ (two-tailed)[35]. Condition labels were randomly permuted 1000 times, following the default method implemented in MNE[36], see "methods".

**Training phase-stimulus-bound representation.** First, we analysed the neural similarity between stimulus-bound representations of the stimuli during the training phase (Fig. 3A, top panel), focusing on both within-domain (Fig. 3B, left panel) and across-domains similarities 3B, right panel).

In both contexts, before and just after task-features presentation, correlation fluctuated around the chance level. Within the spatial context, neural activity was significantly correlated to the one predicted by the stimulus-bound configuration model shortly after stimulus onset for both the SameSt (160–870 ms and 880–1050 ms) and the DiffSt groups (140–1020 ms) (See Supplementary Fig. 3). This pattern was repeated within the conceptual context, where neural activity was similar to the stimulus-bound representations of the stimuli early after stimulus onset for both the SameSt group (80–540 ms, 90–250 ms and 90–320 ms) and the DiffSt group (70–570 ms, 90–580 ms and 560–820 ms) (See Supplementary Fig. 3). All the clusters showed positive correlations.

Interestingly however, in the cross-domain analysis, where the two feature spaces were equated with each other as if they were aligned to a common embedding (Fig. 3B right panel), we found a positive significant cluster in the SameSt group (630–810 ms) (See Fig. 4A) and a trend towards a negative cluster ($p = 0.1$) in the DiffSt group (740–880 ms) (See Supplementary Fig. 3). The positive correlation here indicates that the more aligned the two configurations were in the hypothetical common or abstract cross-domain space (as if the

# Cross-Domain Space

## A Same Structure - Stimulus-Bound representation

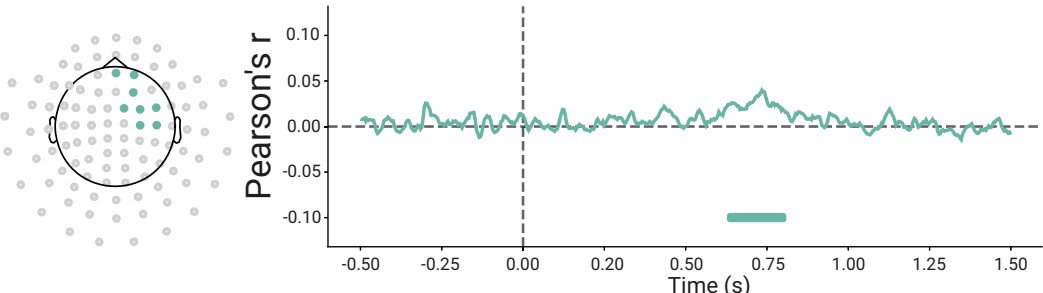

## B Different Structure - Task-Relevant representation

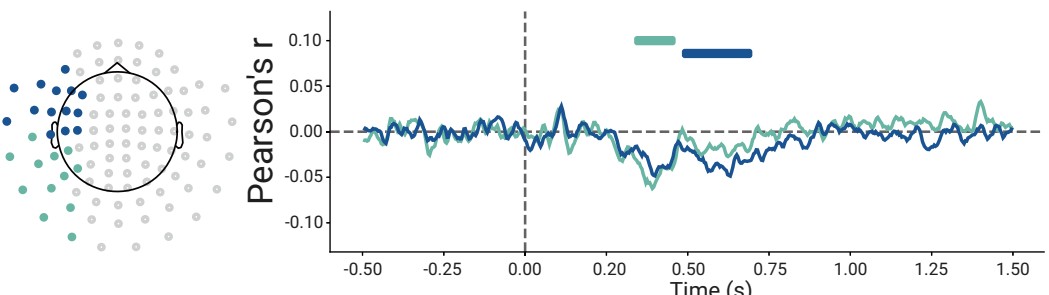

**Fig. 4 | Cross-domain representations during training. A** shows the results of the cluster-permutation correction of stimulus-bound representations for cross-domain space during training for participants in the SameSt group. Clusters are color-coded so that significant sensors, their significant time course and correlation are shown. Each cluster is associated with one color. **B** shows the results of the cluster-permutation correction of task-relevant representations for cross-domain space during training for participants in the DiffSt group. Clusters are color-coded so that significant sensors, their significant time course and correlation are shown. Each cluster is associated with one color.

two feature sets were identical), the more similar the neural activity patterns they elicited. Conversely, the negative correlation indicates that the more aligned the two configurations were in such abstract cross-domain space, the more diverse or dissimilar their neural activity patterns. Consistent results were found when we divided participants into good and bad performers in both DiffSt and SameSt groups (Supplementary Fig. 7).

To summarize, we observed that patterns of neural activity evoked by stimuli configurations in the SameSt group reflected their relational structure in 2D feature spaces, both when we considered the domains separately as well as when we considered them aligned to a common space. This finding was interesting as the two domains in the SameSt group indeed share a similar structure. Thus, the brain seems to reflect this in its neural activity. In the DiffSt group, on the contrary, we observed that patterns of evoked neural activity were reflecting (positive correlations) stimulus-based models within each domain, but when we compared similarities across domains, we observed a negative correlation, mirroring the representational structure of the two task spaces, which are, indeed, anti-correlated.

**Training phase-task-relevant representation.** Then, we analyzed the neural similarity between task-relevant representations of the stimuli (Fig. 3A bottom panel) for both within-domain and across-domain conditions (Fig. 3B) during the training phase. In all the contexts (See

Supplementary Fig. 4), before and just after task-features presentation, correlation fluctuated around the chance level.

Neural activity patterns recorded from the SameSt group did not significantly correlate with predicted distances in the task-relevant models, suggesting that such compressed representation was not reflected in this group's brain activity. This applied to both the within- and the cross-domain analyses. Conversely, looking at the within-domain analyses, in the DiffSt group, within the spatial domain/task, we found two positive significant clusters (170–540 ms and 250–600 ms) (See Supplementary Fig. 4), indicating that neural activity was more similar for compressed spatial configurations that were close in the task-relevant space. Similarly, within the Conceptual domain/task, we found two positive significant clusters showing that the neural activity was more similar for configurations that were close in the task-relevant space (570–780 ms and 870–1150 ms) (See Supplementary Fig. 4). In the cross-domain analyses, in line with our results in the Stimulus-bound analysis (previous paragraph), we found a negative significant cluster (330–460 ms and 480–700 ms) (See Fig. 4B) in the cross-domain analysis, indicating neural activity was more dissimilar when compressed configurations belonging to the two tasks were close in the cross-domain space in the DiffSt group.

In summary, we showed the emergence of task-relevant representation in the DiffSt group, suggesting the emergence of a low-

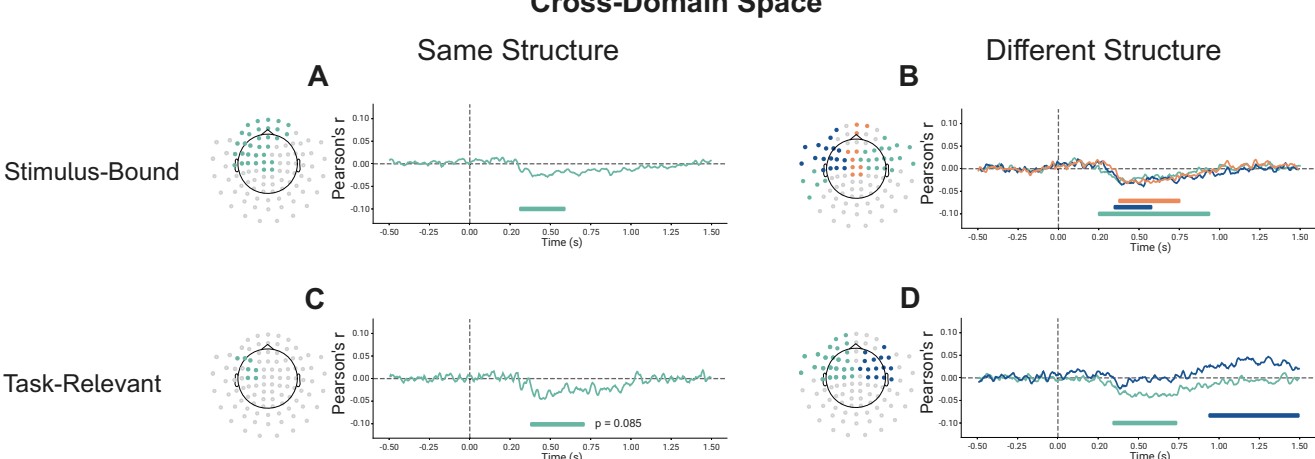

**Fig. 5 | Cross-domain representations during testing sessions. A, B** shows the results of the cluster-permutation correction of stimulus-bound representations for cross-domain space during testing sessions for participants in the SameSt and DiffSt groups. Clusters are color-coded so that significant sensors, their significant time course and correlation are shown. Each cluster is associated with one color. Results are divided into spatial, conceptual and cross-domain spaces for both the SameSt and DiffSt groups. **C, D** shows the results of the cluster-permutation correction of task-relevant representations for cross-domain space during training for participants in the SameSt and DiffSt groups. Clusters are color-coded so that significant sensors, their significant time course and correlation are shown. Statistical significance was determined using a two-sided cluster-permutation test. Each cluster is associated with one color.

dimensional representation. This pattern in neural activity is a potential candidate source of the difference observed in the behavior of the two groups: as participants had, given our design, to differentiate the two domains and focus their attention only on the information "cued" by the seed stimulus, the DiffSt group might have found this operation easier as the task structures facilitated their decorrelation, while for the SameSt group, the shared task structure might have been more detrimental. If this interpretation is correct, then a possible prediction follows: during the test phase, when the behavioral differences between the two groups are attenuated, we should observe these effects to reverse for the SameSt group.

**Testing phase-stimulus-bound representation.** While previous analyses focused on the training phase, we repeated them, now focusing on the testing phase, after learning had happened. In all the contexts (See Supplementary Fig. 5), before and just after task-features presentation, correlation fluctuated around the chance level. In the within-domain analyses, in both the spatial and the conceptual tasks, neural activity was more similar for configurations that were close in the stimulus-bound space. This was significant early after stimulus onset for both the SameSt group (Spatial: 100–1210 ms; Conceptual: 70–1370 ms, 90–460 ms and 520–1000 ms) and the DiffSt group (Spatial: 90–1300 ms; Conceptual: 50–1500 ms and 85–1500 ms).

Crucially, in the cross-domain analysis, we found a significant negative cluster in both the DiffSt group (240–940 ms, 340–590 ms and 370–760 ms) (Fig. 5B) and the SameSt group (300–590 ms) (Fig. 5A), indicating that now in both groups the neural activity was more dissimilar when configurations belonging to the two tasks were close in the cross-domain space. Consistent results were found when we more directly compared training and test sessions in both DiffSt and SameSt groups (Supplementary Fig. 8). Additionally, cross-group validation of significant clusters further supports this pattern, showing that clusters identified in one group exhibit similar effects when tested in the other (Supplementary Fig. 8).

This finding was interesting as the two domains in the SameSt group that share a similar structure might have been detrimental to learning. While during the training the representations of the two tasks were positively correlated, the negative correlation observed now could be a result of the learning process.

**Testing phase-task-relevant representation.** This pattern was also replicated when we analysed Task-relevant representations (Fig. 3A bottom panel). In all the contexts (See Supplementary Fig. 6), before and just after task-features presentation, correlation fluctuated around the chance level. Early after stimulus onset, neural activity in the Spatial task was more similar for configurations that were close in the task-relevant space. This was significant for both the SameSt group (180–1080 ms, 200–1060 ms, 380–710 ms and 480–1020 ms) and the DiffSt group (800–1360 ms and 150–780 ms).

In the Conceptual task, neural activity was more similar for configurations that were close in the task-relevant space. This was significant early after stimulus onset for both the SameSt group (70–1110 ms, 590–1220 ms and 1090–1360 ms) and the DiffSt group (60–1500 ms and 440–1500 ms).

In the cross-domain analysis, we found two significant negative clusters in the DiffSt group (340-740 ms and 930-1500 ms) (Fig. 5D) and a negative trending towards significance cluster in the SameSt group (370–710 ms; $p = 0.085$) (Fig. 5C). The negative correlation means that neural activity was more dissimilar when configurations belonging to the two tasks were close in the cross-domain space.

In summary, we showed the emergence of task-relevant representation in both the DiffSt and SameSt groups, suggesting the emergence of a low-dimensional representation. This pattern in neural activity shows how, during testing, both groups were able to differentiate the two domains, possibly facilitated by the decorrelation of their representations.

**Artificial neural networks offer insight into the difference in learning performance**

To understand the difference in learning speed between participants who learned either the same or different structure conditions, we trained neural networks to perform tasks that are analogous to the experimental tasks (Fig. 6a, b). The networks were given input variables relevant to each task as well as input variables that signaled the appropriate context for a given trial (similar to the seed identity in the experimental task). Then, some networks were trained to perform tasks with the same structure across the conceptual and spatial domains (Fig. 6b, top) and other networks were trained to perform tasks that had different structures across the two domains (Fig. 6b,

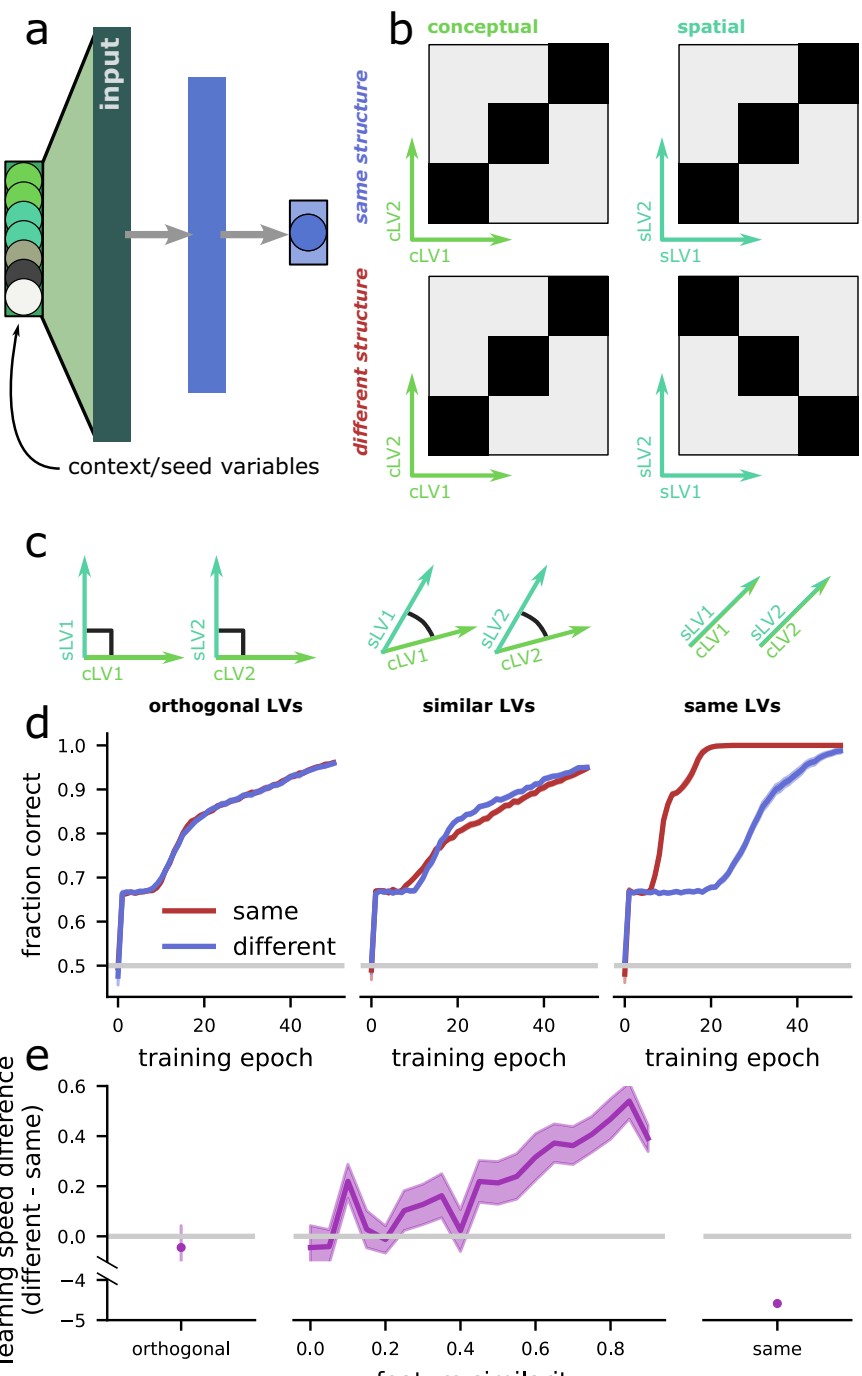

**Fig. 6 | Schematic and learning performance of the neural network across different input structures and tasks. a** Schematic of the neural network. There are two latent variables (LVs) relevant to the first context (cLV1 and cLV2), two relevant to the second (sLV1, sLV2), two irrelevant variables, and two variables that signal the context (i.e., the seed identity in the experiment). There is a single hidden layer and one output unit. **b** The tasks the network is trained to perform in two contexts (conceptual and spatial, left and right). The tasks are a coarser version of the experimental task and also have the same structure and different structure

conditions. **c** Schematic of the three input conditions for the network: (left) the features are all orthogonal to each other; (middle) pairs of features across the two contexts are encoded along semi-orthogonal directions; (right) there are only two, rather than four, total features. **d** The learning trajectory for each input structure, averaged over $n = 50$ repetitions for each condition. **e** The average total difference in learning trajectory across the networks above for orthogonal inputs (left), a range of semi-orthogonal inputs (middle), and identical inputs (right). Error bars represent the standard error of the mean.

bottom). More details about the modeling setup are given in Artificial neural network modeling in Methods.

We began by providing the input variables to the network in perfectly orthogonal dimensions (e.g., each variable was a different "input" to the network; Fig. 6c, left). This leads to identical learning speed across the networks trained with the same- and different structure (Fig. 6d, e

left). This identical learning speed emerges because the same and different structures are not distinguishable from the perspective of the network. Since the variables are all in orthogonal subspaces, there is no prior relationship between the variables in the first and second contexts.

How can we induce an appropriate bias in the network? Instead of providing the task features along orthogonal dimensions, we provide

them along semi-orthogonal—or, simila—dimensions (Fig. 6c, middle), where we index similarity by the cosine similarity between the vectors defining each dimension. This means, for example, that the amount of water in context 1 will be related to the x-position of the seeds in context 2. This manipulation will bias the network toward treating the different features similarly and produce interference between the two features that must be overcome for reliable task performance. This manipulation causes faster learning for tasks with different relative to the same structure (Fig. 6d, e, middle) and qualitatively replicates the behavioral effect observed in our participants.

Finally, we explore the case where the network is only given two task-relevant features (Fig. 6c, right). This is an extreme version of the similar feature case above, where the features are identical (i.e., perfectly aligned, or with zero angle). In this case, the same structure task is no longer necessarily contextual, since the network (or subject) can ignore the contextual variable (or seed) and still perform the task perfectly. This is not the case for the different structure task, where context remains relevant. The networks trained with the same structure learn far faster than those trained with different structure tasks (Fig. 6d, e, right).

Overall, the neural networks reveal the conditions in which interference between features across the same and different contexts appears. When the relevant features are represented in initially non-orthogonal subspaces, then both the same- and different-structures require them to be orthogonalized for reliable task performance, but the same-structure orthogonalizes more slowly. The neural network makes predictions that can be tested in further work, where feature similarity is manipulated—as well as that can be tested by more granular neuro-imaging methods, where we could test the central prediction of the model: That at the start of training, the features will be represented in positively correlated subspaces. The slower orthogonalization predicted by the model is also born out by the data, where the MEG signal from the training period is positively correlated in same-structure participants, but negatively correlated in the different-structure participants.

## Discussion

We studied the temporal dynamics of when and how the brain acquires and shares representations of and across tasks and how sharing these structures affects behavior. Participants learned two tasks where a low-dimensional manifold defined the associations between stimuli and outcome (representation), which were either similar between the two tasks or not (see Fig. 1). The brain can benefit from the similarities between tasks to facilitate efficient learning. We expected shared representations to improve performance during learning. However, our behavioural analyses showed a surprising finding: during training, tasks sharing a common low-dimensional structure exhibited interference, leading to worse learning performance (see Fig. 2A). Nonetheless, this interference decreased with practice, with performance differences disappearing during the testing sessions. Notably, shared representations did not impact generalization (see Fig. 2B).

Through multivariate analyses of MEG data, we explored the emergence and impact of shared structures across different dimensionality and learning stages. We have identified a two-dimensional representation of task stimuli within-domain emerging shortly after stimulus onset (see Supplementary Figs. 3 and 5). Notably, during the training phase, in the DiffSt group, we found evidence of a compressed representation supporting the hypothesis of the emergence of a task-relevant representation (see Supplementary Fig. 3). Although the stimulus-bound and task-relevant models are correlated, their neural correlates revealed distinct patterns, indicating that they capture different aspects of the representational structure. However, such representation was not observed in the SameSt group, possibly because of differences in performance (See also refs. 25,37). Crucially, participants in the SameSt group represented stimuli from both tasks

more similarly in the cross-domain space during training (Fig. 4A), suggesting a shared representation that may have interfered with task performance. This positive correlation between the two tasks aligns with the task structure they were given, where the underlying low-dimensional representations were the same, potentially leading to confusion or interference when trying to distinguish between them. Conversely, the DiffSt group exhibited dissimilar representations, meaning that neural activity patterns associated with the two tasks were more distinct (Fig. 4B). This negative correlation might reflect the orthogonal nature of the task structures in this group, reinforcing the idea that the brain encoded each task in separate neural substrates. This distinct encoding likely helped minimize interference, leading to more stable task performance. During the testing phase, representation similarity shifted for the SameSt group (Fig. 5A–C), suggesting independent encoding of tasks in neural subspaces, coinciding with improved behavioural performance. Moreover, we observed that the sensors involved in the representations shifted over the course of learning, initially engaging more sensory-related sensors during training and stimulus-bound representations, transitioning toward more frontal areas during the test phase and for between-task correlations. This shift aligns with the idea that learning drives neural representations from perceptual encoding toward higher-order cognitive regions, potentially reflecting increased cognitive control, task separation and multi-tasking[26,38–41]. In our study, we only assessed sensor-level data; therefore, interpretations regarding the underlying anatomical sources are based on the standard MEG sensor layout and the spatial distribution of significant clusters, and should be taken with caution.

The neural network model trained to perform a similar task provided us with further insight into the nature of the shared representations (See Fig. 6d). When the two tasks had orthogonal dimensions, the structure of the tasks did not affect the network's performance, as the same and different conditions, from the network's perspective, were effectively identical. However, when the network was trained on tasks with identical dimensions, it learned faster when the structures were similar, effectively treating them as a single task. Interestingly, when the dimensions of the two tasks were similar but not identical, the neural network, much like the human brain, experienced interference when attempting to use overlapping representations for these tasks. These results suggest that the brain might compute the dimensions related to different tasks in distinct ways. However, the low-dimensional similarity between the task structures affects performance, likely due to the cognitive interference that arises from attempting to apply overlapping representations. This implies that while the brain can efficiently share representations across highly similar tasks, it must carefully manage partially similar tasks to avoid detrimental interference. Our network findings align with recent work by Holton et al.[42], which demonstrated that both artificial and biological learners face a trade-off between transfer and interference when learning multiple tasks. Similar to our results, their study showed that for dissimilar tasks, networks encoded each task in separate subspaces to avoid interference, whereas for similar tasks, they relied on overlapping representations. In contrast to our findings, they observed that reusing existing representations for similar tasks accelerates learning but also causes interference with prior knowledge. This discrepancy may stem from the different training regimes used: while they employed a blocked training regime between tasks, we used an interleaved one. Future studies could model the temporal evolution of neural representations in MEG. Similar approaches have been explored in vision[43] using artificial neural networks, and future work could extend our framework to capture these dynamics in cognitive tasks more effectively.

More broadly, our results speak to cognitive control theories, which suggest that while shared representations can facilitate generalization and the acquisition of novel tasks, they also introduce the

potential for interference and associated costs[31,32]. However, unlike classic cognitive control and interference paradigms such as the Stroop task[44,45], where incongruencies between task features are inherent to the perceptual features and task demands, in our study, the interference emerged from learning the underlying structure that links features to outcomes. This distinction is critical, as it highlights a different mechanism of interference-one that arises dynamically through the similarity in integration of otherwise unrelated, task-relevant features during learning rather than from predefined task demands. Consistent with prior work showing that practice can mitigate interference, we observed a gradual shift in neural representations: the correlated activity in the SameSt group transitioned toward an anticorrelated pattern similar to that of the DiffSt group. This adaptation aligns with theories suggesting that the brain resolves interference by progressively separating task representations through mechanisms such as pattern separation and encoding them in distinct neural subspaces[46–54].

Notably, training participants with an interleaved regime might have contributed to the observed interference. A blocked training regime, where participants focus on one task at a time, might mitigate interference and boost learning and transfer by facilitating neural task separation (see ref. [42]). Future studies comparing different training regimes on shared representation dynamics could provide insight into critical stages of transfer learning. It may be necessary for participants to first learn one task before introducing a second one, leveraging shared representations to facilitate novel task acquisition[23,50,55–59].

More broadly, our findings align with the idea that task representations change their structure over the course of learning and experience, gradually shifting toward increasingly task-tailored representations that balance generalizability and separability[33]. This transformation is particularly relevant in the transition from cognitively controlled to automatically controlled behavior. One perspective suggests that automaticity emerges as control shifts from managing compositional codes, initially supporting generalization but inefficient for multitasking, to more efficient conjunctive codes that integrate task-relevant information[33].

Finally, recent studies have pointed to the possibility that task representations resemble cognitive maps[60–62]. Representations are structured within relational maps, with relevant features determining tasks' relative positions[63–66]. Distances in these maps code for similarity, placing similar tasks closer than dissimilar ones[67–70]. Our findings show the brain's ability to organize and manipulate task-related information similar to a cognitive map, with shared representations potentially acting as points of convergence or overlap in this map.

## Methods

### Participants
A total of 60 volunteers participated in the experiment (mean age = 27.91, SD = 4.21, 27 males, 33 females; Sex was self-reported by participants and was not considered as a factor in the analyses). All the participants were naive to the purpose of the experiment. Data from six participants were discarded because their performance was below the chance level (56%) in both the spatial and the conceptual tasks. We performed analyses on the remaining 54 participants. All participants gave informed written consent, and the study procedure was approved by the local institutional review board of the University of Leipzig (Ethics Reference Number: 045/22-ek). At the end of the experiment, participants received reimbursement for their participation.

### Apparatus and stimuli
Participants completed the MEG experiments inside a sound-attenuated, dimly lit, and magnetically shielded room. Stimuli were displayed on a rear-projection screen with a spatial resolution of 800 × 600 pixels and a refresh rate of 60 Hz using the Psychophysics Toolbox (http://psychtoolbox.org/)[71] for Matlab (Mathworks). It's worth noting that the projector was changed after subject 28, although the proportions remained consistent, and the refresh rate was increased to 120 Hz. Two images of seeds were used as cues to which kind of task (spatial or conceptual tasks) participants had to do (See the left panel of Fig. 1). In the conceptual task, the opacity of a sun and a drop of water images governed the simulated amount of water and light received by the seed. Participants were instructed that greater opacity indicated higher levels of water or sunlight. Literature suggests that lower magnitudes are represented to the left and downward, while higher magnitudes are represented to the right and upward[72–74]. The spatial task involved a black dot positioned on a white square, with the vertical and horizontal placements of the dot dictating the virtual planting location for the seed. We sampled a subset of feature combinations from these two-dimensional spaces to create the stimuli; see Supplementary Fig. 1 for a breakdown of all the feature combinations that were used to create the stimuli.

**Task structures.** We used two different feature-outcome maps, which we refer to as structures (see Fig. 1). These structures can be approximated by two different "diagonal" rules, allowing participants to learn the value of different spatial or conceptual combinations without sampling them first. Given the large number of possible stimulus-outcome associations, learning the task without recognizing this underlying structure would be nearly impossible. Thus, this drives the discovery of a lower-dimensional, compressed representation of the task structure to facilitate learning. Both task structures were defined using a deterministic mapping where the rounded value of the log-odds of the outcome was a quadratic function of stimulus characteristics, either the opacity of the two images or the position in space.

$$\log\left[\frac{p(y_t=1)}{p(y_t=0)}\right] = (u_t - \mu)^T W(u_t - \mu) + w_0$$
$$W = 2.4 \times \begin{bmatrix} -0.71 & w_d \\ w_d & -0.71 \end{bmatrix}$$
$$\mu = [50.5, 50.5]^T \qquad (1)$$
$$w_0 = 500$$
$$u_t = [Feature\,1, Feature\,2]^T$$

$u_t$ is a matrix consisting of $n$ columns, each representing a unique combination of features, and two rows containing values scaled from 1 to 100. $t$ is indexing all the possible combinations of features. These parameters are the feature values Water and light opacity for the conceptual task, and x and y positions for the spatial task. Flipping the sign of the wd parameter in this mapping produced the two rotated and orthogonal structures depicted in Fig. 1. The parameters 2.4 and 0.71 have been arbitrarily chosen to create the maps.

### Experimental design
Each participant was trained to do two tasks in an interleaved fashion. In the first one, the conceptual task, they had to learn the association between the opacity of a sun and a drop of water images and an outcome (grow or die); in the second one, they had to learn the association between a position in space and an outcome (grow or die). Importantly, spatial positions did not affect the conceptual task, and conceptual features did not affect the spatial one.

Participants were randomly assigned to either a "Same structure" (SameSt) or a "Different structure" group (DiffSt), with 30 participants in each group (27 per group after discarding bad performers). The features-outcome mappings for the SameSt group were generated using $wd = 0.71$ for both spatial and conceptual tasks (See previous paragraph and equation (1). The features-outcome mappings for the DiffSt group were generated using $wd = 0.71$ for the conceptual task and $wd = -0.71$ for the spatial task.

## Procedure

The experiment comprises four distinct phases, as illustrated in the right panel of Fig. 1 and lasted about 2 h. Throughout these phases, participants engage in a computerized task where they assume the roles of scientists within a biology laboratory. In this simulated scenario, the laboratory has successfully cultivated two novel plant species, and participants are tasked with acquiring knowledge about the specific environmental conditions required for the successful germination of these plant seeds.

**Training.** As illustrated on the right part of Fig. 1, during the first phase, the training task, in each trial, participants will first be prompted with one of two seeds cueing which task to focus on and then the stimuli appeared and stayed on screen for 2500 ms maximum or until a response was made. Responses were made on a standard button box, one button indicating a prediction of 'grow' and another predicting "die". Responses not given within the required time constitute "missed trials". Right after the button press, feedback was provided, saying "correct" if the prediction was correct, "incorrect" if it was not and "too slow" if they missed the trial (no response within 2500 ms). By making predictions and receiving feedback, participants learned the association between relevant features and the outcome "grow" or "die" as in a classic associative learning task. During the training phase, participants were exposed to 18 different configurations in both spatial and conceptual tasks. Each configuration was repeated 8 times, with a different irrelevant configuration. For example, in the spatial condition, each spatial configuration was paired with 8 conceptual configurations (4 associated with "grow" and 4 with "die"). The total number of trials in the training phase was thus 288 (18 configurations × 8 repetitions x 2 tasks). It's important to note that some stimuli (i.e., coordinates in the two-dimensional feature spaces) were common between the two contexts, while others were unique to each context, providing a multifaceted learning experience for the participants.

**Test.** During the second phase, the test tasks, we tested participants' knowledge about the structures learned during the training. We presented old and new stimuli to test for differences in memorization, transfer and generalization. Some of the new stimuli were new in both contexts, and some were previously encountered (old) in one context but entirely new in the other context. During the testing phase, participants were exposed to 28 different configurations in both spatial and conceptual tasks. Like during the training session, each configuration was repeated 8 times, with a different irrelevant configuration. For example, in the spatial condition, each spatial configuration was paired with 8 conceptual configurations (4 associated with "grow" and 4 with "die"). The total number of trials in the training phase was thus 896 (28 configurations × 8 repetitions x 2 tasks). This phase is divided into "first testing session" and "second testing session" with 8 repetitions each, separated by a navigation task that will be analysed separately.

**Navigation.** The navigation task addresses a distinct question that will be analyzed separately. Here we provide a short description of the task. Participants were presented with one of the two seeds and an expected outcome: "grow" or "die". Participants were then asked to navigate the appropriate (spatial or conceptual) space, by varying the two features (x and y for the spatial seed, amount of sun and rain for the conceptual seed), locate a position on the map and "plant" the seed according to the outcome requested.

Even though data from the navigation task have not been analysed as they pertain to a different question, we show how participants' general performance does not change between testing sessions.

## MEG acquisition and preprocessing

Neuromagnetic data were acquired using a 306-sensor MEG system (204 planar gradiometers, 102 magnetometers, MEGIN Vectorview system) at the Max Planck Institute for Human Cognitive and Brain Sciences, Leipzig, Germany. MEG data were recorded at 1000 Hz sampling frequency. The MEG data were preprocessed offline using MNE Python software[36]. Specifically, we low-pass and high-pass filtered the data at 0.5 and 100 Hz. Notch filtering at 50 Hz was performed to remove the power line artifact. Data across different runs were aligned to an individual common head position. The Maxwell filter was then applied. We epoched the data from 500 ms before the onset of the stimulus to 1.5 s following it. We visually inspected the epochs to identify and exclude epochs with artifacts, including those with evident eye movements, which were discarded to avoid potential contamination of neural signals. However, as we did not record eye tracking data, a more detailed analysis of eye movements was not possible. On average, over the different phases, 4% of the trials were excluded. We never discarded more than 10% of the trials. Independent component analysis was then performed in each subject to remove eye movement and artifact components, and the remaining components were then back-projected to channel space. No trials were discarded during this procedure. A maximum of 2 non-neighbouring sensors were interpolated per participant. We interpolated 2 sensors for 4 participants, 1 sensor for 3 participants and none for the remaining 47. Before computing Event-Related Fields (ERFs), we applied baseline correction based on the activity pre-stimulus onset, applied a low-pass filter at 30 Hz and resampled the data at 250 Hz.

## MEG data analysis

We performed univariate and multivariate analyses of the data to get a deeper insight into the relationship between stimulus-bound as well as task-relevant representations and participants' performance. First, we performed representational similarity analysis using a searchlight at the sensor level to investigate the neural chronometry of the task representations created and how it was related to performance. Second, we computed ERFs to investigate differences between the groups in generalization.

**Representational similarity analysis.** Our experiment used different stimulus configurations, each being a unique combination of either conceptual or spatial features. 18 configurations, per task, were used during training and 28 during test. Each configuration was repeated multiple times (8 times in training, 8 times in test), each configuration was paired with 8 different irrelevant configurations (4 mapped to "grow", 4 to "die"). For example, one spatial configuration was paired with 8 different conceptual configurations, 4 associated with grow and 4 with die outcomes. These pairings were counterbalanced to prevent systematic associations. Here, we used RSA to identify the relationship between these configurations and the multivariate MEG (gradiometers) signals as they evolve over time[34]. We selected the peristimulus signal from −500 ms to 1500 ms with respect to stimulus onset.

*Model matrices:* We generated six distinct model matrices utilizing Euclidean distances. These models fall into two main categories: First, the Stimulus-bound Model, the stimulus-bound Model, reflects distances within each of the two-dimensional spaces defined by screen features like the opacity of sun and water or position (x and y coordinates). Second, the Task-relevant Representation involves computing distances along a single dimension computed based on task relevance, by subtracting feature 2 from feature 1, offering insights into the core aspects of the task structure. By measuring distances in this way, we can ensure that the representation reflects task-specific features rather than motor or spatial biases. These models were constructed for both spatial and conceptual domains, as well as by calculating distances between the conceptual and spatial task configurations in a cross-domain space. The stimulus-bound and task-relevant RDM matrices were correlated, with base correlations of 0.6

during training and 0.5 during test in the Conceptual task and 0.42 during training and 0.5 during test in the Spatial task.

*Empirical neural dissimilarity matrices*: The empirical matrix represents the neural dissimilarity matrix derived from MEG data. Specifically, we use correlation distance (1-Pearson's r) to compute dissimilarities between neural response patterns, consistent with our prior work using similar task structures[25,27]. The empirical dissimilarity matrix was constructed by adopting a searchlight approach at the sensor level. This method involved calculating the averaged neural response for each configuration. Neural dissimilarity matrices were then computed separately for each task (spatial and conceptual) and based on averages across trials for each unique task configuration, which allows us to reduce noise and average out irrelevant feature variation. This averaging step is crucial as it allows us to isolate distances along the relevant task dimension while minimizing the influence of irrelevant dimensions. For each sensor, we selected neighbouring channels within a maximum distance of 40 mm and computed the correlational distance between all the configurations.

The configurations in the model and empirical matrices are organized such that each row and column corresponds to a specific stimulus configuration, with the values indicating the distance (Euclidean for the model and correlational for the empirical) between pairs of configurations. Across all time points and sensors, we conducted a correlation (Spearman) analysis between the neural dissimilarity matrix and the model matrices. Statistical significance was assessed non-parametrically at the group level using a cluster-based permutation approach with a cluster-forming threshold of $p < 0.05$ (two-tailed), and a corrected significance level of $p < 0.05$ (two-tailed)[35]. Condition labels were randomly permuted 1000 times, following the default method implemented in MNE. This provides an automatic method for finding significant clusters, corrected for multiple comparisons, that does not depend on a priori selection of time window and channels.

### Artificial neural network modeling

**Model architecture.** We trained feedforward neural network models with a single hidden layer to perform an analogous task to the one used in the behavioral experiments. The neural network had one hundred units in the hidden layer, each with a ReLU activation function. The network had a single output unit, used to signal the category of the stimulus (e.g., whether the stimulus would grow or die in the particular position and conditions shown, as in the experiment). The input to the network was composed of six latent variables, corresponding to the spatial features, the conceptual features, and the context (e.g., the seed identity). The spatial and conceptual feature latent variables could each take on three values. The two contextual variables were constrained to be one-hot. The network received a linearly transformed version of these input variables, $x = Mz$, where $z$ are the latent variables, $M$ is a $6 \times 6$ matrix, and $x$ are the inputs to the network. The columns of $M$ were selected with correlation structure between the corresponding spatial and conceptual latent variables, such that,

$$\frac{M_1 \cdot M_3}{||M_1||_2 ||M_3||_2} = \frac{M_2 \cdot M_4}{||M_2||_2 ||M_4||_2} = c \qquad (2)$$

where $M_i$ is the $i$th column of $M$, $||M_i||_2$ is the length of the $i$th column of $M$, and $c$ is the desired level of feature similarity.

**Optimization.** The network was trained for 50 epochs, with a batch size of 100 training examples drawn from a set of 2000 total examples. The parameters were optimized with the Adam optimizer with a learning rate $10^{-3}$.

### Reporting summary

Further information on research design is available in the Nature Portfolio Reporting Summary linked to this article.

## Data availability

Raw data are protected and are not available due to data privacy. Preprocessed data will be made available upon request to the corresponding author. Processed data and source data are provided with this paper and available at https://doi.org/10.6084/m9.figshare.30353515. Source data are provided with this paper.

## Code availability

The code for the Neural Network was written in python, using tensorflow[75] and numpy[76]. The code is freely available on github.

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

## Acknowledgements

We thank Yvonne Wolf-Rosier for her invaluable assistance with participant recruitment and data collection, Lola Kuhn for her support with recruitment, Kerstin Schumer for her exceptional management of the project. M.H. is supported by the Max Planck School of Cognition. C.F.D.'s research is supported by the Max Planck Society, the European Research Council (ERC-CoG GEOCOG 724836), the Kavli Foundation, the Jebsen Foundation, the Centre of Excellence scheme of the Research Council of Norway-Centre for Neural Computation (223262/F50), the Egil and Pauline Braathen and Fred Kavli Centre for Cortical Microcircuits, and the National Infrastructure scheme of the Research Council of Norway-NORBRAIN (197467/F50).

## Author contributions

N.M., S.V., and C.F.D. conceived the study. N.M. programmed the tasks and collected the data with support from M.H.; N.M. analyzed the behavioral data with assistance from M.H.; N.M. preprocessed and analyzed the MEG data with assistance from B.M.; J.J. and S.F. designed and analyzed the artificial neural network. N.M. wrote the manuscript with input from S.V., C.F.D., J.J., S.F., B.M., and M.H.; J.J. and S.F. wrote the neural network section. C.F.D. acquired the funding.

## Funding

## Competing interests

The authors declare no competing interests.
