## [Transparent Peer Review file · Nature Communications]

The effects of task similarity during representation learning in brains and neural networks

Corresponding Author: Dr Nicholas Menghi

Version 0:

Reviewer comments:

Reviewer #1

(Remarks to the Author)

The authors present a manuscript aimed to characterize the similarity of task encoding with neural representations throughout learning and generalization. The authors design a simple experiment aimed to manipulate the degree of interference in the task encoding/presentation, where one variant of the task has two features that are correlated (conceptual and spatial features), and the other task encodes these features independently. They trained/evaluated two separate cohorts of human participants on each of these variants, evaluating their training and testing performances. Overall, they found that humans presented with the non-interfering task variant (i.e., independent features) tended to perform better than those trained on the interfering task variant. Subsequent neural analyses (via RSA on MEG data) revealed differences across the groups. Experiments with computational models revealed an overall similar pattern, whereby artificial nets trained with correlated task features tended to learn more slowly than those trained with independent task features.

Overall, I encountered several major issues with the paper. 1) Novelty: Much the conclusions and findings from this paper are consistent with those from the human cognitive control literature, namely studies using interference tasks such as the Stroop task (Cohen et al., 1990; Freund et al., 2021). 2) Clarity of the findings (particularly the RSA results): I found that many of the findings were not as clear nor obvious as to how they were presented in the Abstract and the Introduction. 3) Utility of the model: It seems to me the model was not necessary, and that this task can be performed by a linear classifier. Below, I explain these issues in detail.

Major

1. Much of these results do not seem particularly novel over and above the prior literature. It is known that incongruent conditions in task interference tasks (e.g., incongruent v. congruent conditions within the Stroop task) are harder to learn, but performance differences can be attenuated with practice (Davidson et al., 2003). Moreover, other studies have also evaluated the effects of neural representations across practice/training in humans (fMRI/EEG/MEG) and models (Cole et al., 2011; Dekker et al., 2022; Flesch et al., 2023; Freund et al., 2021; Ito et al., 2022; Kikumoto et al., 2024). I recognize that while the authors referenced some of these papers, exactly what insights were uniquely afforded from their paper relative to these was not entirely clear.

2. I was not entirely clear on the main findings from the RSA analysis in MEG data. The abstract says “Crucially, practice reduced this performance gap and shifted the neural representations of the tasks in the same-structure group towards anticorrelation, like those in the different-structure group.” Unless I missed it, I did not find an analysis that supported this claim. More broadly, it was hard to compare the RSA time course traces, given that the cluster time courses were sourced from the different MEG source areas. Moreover, no effort was seemed to try and interpret/understand the sources from the MEG data.

3. The narrative surrounding the behavioral results seem a bit strong, given that the only difference found in the training performance between the two groups, and not during test time. Was there any difference in the RT across the two conditions?

4. Utility of the model: I found the model experiments to be generally unsurprising. It seems as though the difference in the results could be explained by simply measuring the dimensionality/entropy of the input space. In other words, the greater the dimensionality of the input encoding, the easier the model would learn the task, which seems intuitive. (In essence,

assuming two separate groups/models for the in two conditions (same v. diff), wouldn't the task only require a linear classifier for each group? Is an ANN truly necessary?)

5. Related to the above, by 50 repetitions per model equivalent to 50 random seeds? Or the number of epochs? Why are there no error bars in figure 6d?

6. I found the MEG RSA results to be a hard to make sense of. How should one interpret negative v. positive correlations with the ground truth RSA matrix? Are the results dependent on choice of distance metric (of both the RDM matrix and/or the difference between neural and task RDMs)? How do I interpret the very low effect sizes of the actual correlation traces (Fig. 4, 5)?

Minor

1. How do the authors interpret the location of the clusters in the MEG source, in figures 4 and 5?

2. The discussion of learning a compressed representation that is optimal for learning the task structure seems a bit like an overgeneralization, given that the compression is from a 2D encoding of space to a 1D encoding of relative differences to the diagonal.

3. What was the base correlation between the stimulus-bound v. task-relevant RDM matrix?

References

- Cohen, J.D., Dunbar, K., McClelland, J.L., 1990. On the control of automatic processes: a parallel distributed processing account of the Stroop effect. *Psychol. Rev.* 97, 332–61. <https://doi.org/10.1037/0033-295X.97.3.332>
- Cole, M.W., Etzel, J.A., Zacks, J.M., Schneider, W., Braver, T.S., 2011. Rapid Transfer of Abstract Rules to Novel Contexts in Human Lateral Prefrontal Cortex. *Front. Hum. Neurosci.* 5, 142. <https://doi.org/10.3389/fnhum.2011.00142>
- Davidson, D.J., Zacks, R.T., Williams, C.C., 2003. Stroop Interference, Practice, and Aging. *Neuropsychol. Dev. Cogn. B Aging Neuropsychol. Cogn.* 10, 85–98. <https://doi.org/10.1076/anec.10.2.85.14463>
- Dekker, R.B., Otto, F., Summerfield, C., 2022. Curriculum learning for human compositional generalization. *Proc. Natl. Acad. Sci.* 119, e2205582119. <https://doi.org/10.1073/pnas.2205582119>
- Flesch, T., Nagy, D.G., Saxe, A., Summerfield, C., 2023. Modelling continual learning in humans with Hebbian context gating and exponentially decaying task signals. *PLOS Comput. Biol.* 19, e1010808. <https://doi.org/10.1371/journal.pcbi.1010808>
- Freund, M.C., Bugg, J.M., Braver, T.S., 2021. A Representational Similarity Analysis of Cognitive Control during Color-Word Stroop. *J. Neurosci.* 41, 7388–7402. <https://doi.org/10.1523/JNEUROSCI.2956-20.2021>
- Ito, T., Klinger, T., Schultz, D.H., Murray, J.D., Cole, M., Rigotti, M., 2022. Compositional generalization through abstract representations in human and artificial neural networks. *Adv. Neural Inf. Process. Syst.*
- Kikumoto, A., Shibata, K., Nishio, T., Badre, D., 2024. Practice Reshapes the Geometry and Dynamics of Task-tailored Representations. *bioRxiv* 2024.09.12.612718. <https://doi.org/10.1101/2024.09.12.612718>

(Remarks on code availability)

Reviewer #2

(Remarks to the Author)

Summary

This study investigates whether a shared latent structure across two tasks facilitates learning compared to tasks with distinct latent structures. The authors used MEG to measure neural representations in 54 human participants assigned to two experimental groups: one with a shared task representation and the other with a different representation. Behavioural analyses revealed an unexpected result - contrary to the authors' predictions, participants exposed to two different latent structures learned more slowly than those with distinct task structures. Representational similarity analysis (RSA) further indicated that, irrespective of whether the decision boundary was shared, participants in both groups constructed negatively correlated representations of task feature spaces, as well as the decision boundaries (grow vs. die) across tasks.

This study presents a valuable opportunity to uncover key computational mechanisms underlying the ability to utilise abstract representations in humans. It is particularly insightful in examining how partial overlap in representation - specifically in the decision boundary but not in the stimulus space - affects learning. A notable strength of the study is also the inclusion of two experimental groups, enabling direct comparisons of learning effects.

However, the manuscript lacks critical details regarding the analyses and the rationale behind key methodological decisions, which significantly hinders a thorough assessment of the reported findings. Additionally, the operationalisation of task structure as a means to test the facilitatory effects of shared representation spaces appears to be an overextension. The results suggest that the study is less about the advantages of a shared feature space and more about the challenges posed by shared decision boundaries.

While my concerns are substantial, I believe this study has the potential to make a meaningful contribution. I recommend the manuscript for publication, provided that the authors adequately address these issues.

Major concerns:

The analysis lacks crucial methodological details and a clear rationale for key decisions. For instance, was neural activity averaged across trials, or were representational matrices computed from single-trial data? What was the number of trials per unique condition, and how many unique conditions were included? Additionally, not all the RSA matrices are being shown in detail and what is being presented lacks important details for interpretation (e.g., axis labels). It is also unclear why neural activity was only analysed during the period when spatial or conceptual features were presented, and the seed-locked period was not explored. These omissions hinder the transparency and interpretability of the findings. I have outlined further specific concerns alongside the corresponding issues below.

1. It remains unclear how participants in the SameSt group could have constructed a shared feature representation between the two tasks. While the geometry and orientation of the decision boundary were preserved across tasks, the feature spaces should have remained orthogonal to prevent irrelevant features from influencing the processing of relevant ones (i.e., using the space feature map to solve a conceptual task). Consequently, the task design in this group does not inherently promote shared task structure but rather a shared decision readout. Referring to this condition as "same structure" in the introduction appears misleading. A task could more justifiably be described as having a "shared task structure" if the feature spaces were correlated at the single-trial level, as this could yield an explicit behavioural advantage (e.g., allowing participants to ignore one of the feature spaces). My comment is based on two key assumptions: (1) that both the spatial map and conceptual pictograms were presented simultaneously, and (2) that the values of the task-irrelevant feature space were randomly assigned on each trial. There is a mention in the Methods section that the feature spaces "were not related," and Figure 1 suggests that they were presented simultaneously. However, these details should be explicitly stated in the text to ensure clarity.

2. Figure 3A suggests that task-relevant representations are measured as an interaction between the binary decision boundary (Grow vs. Die) and the feature space (spatial or conceptual). What is the rationale behind this approach? Why is it relevant to represent a "Die" differently if it was based on the opposite side of the feature map? Why not decode the binary decision boundary alone, given that this is the element shared in the SameSt group? As noted earlier, it seems unlikely that participants would develop a shared representation of the feature space, as doing so would likely impair performance. Perhaps there is a justification for using this interaction model, but if so, it needs to be explicitly stated.

3. As currently described, the text suggests a potential confound between the task-relevant representation shared across domains and motor preparation. As mentioned earlier, the shared (across-domain) task-relevant representation appears to capture a combination of the decision boundary (Die vs. Grow) and the feature space used across tasks. The Die vs. Grow boundary directly corresponds to the two response buttons (e.g., left vs. right). Was the button-to-decision mapping randomised throughout the experiment? This issue needs to be explicitly acknowledged, and the procedures used to mitigate it should be detailed. If randomisation was not implemented, all "across-domain" analyses based on the feature-locked period are likely capturing motor preparation signals rather than genuine task-related representations.

4. What is the rationale for using a searchlight procedure, and what does the procedure entail in detail? Specifically, how was the cross-validation implemented within the searchlight protocol, and how were the statistical tests corrected for multiple comparisons? Additionally, how did the results compare when the analysis was conducted simultaneously across all sensors? I recommend providing a more detailed description of these methodological choices, along with justifications for each decision.

5. The procedure for computing distance/correlation matrices remains unclear. The analysis begins with the computation of an observed (neural) similarity matrix - presumably equivalent to a correlation matrix? However, Figure 3B presents $r-1$ values, which raises questions. Why is $r-1$ shown instead of raw correlation values (I recommend plotting both correlation and distance matrices in colour with appropriate colour bars for better clarity)? Additionally, is the observed correlation matrix computed between condition averages? How many unique conditions are there? Are "configurations" considered unique conditions? Furthermore, is this process performed separately for each task (spatial and conceptual)? This needs to be explicitly stated and all RSA model matrices need to be shown. I also suggest using Mahalanobis distance instead of correlations for the observed matrices (with covariance estimated from the data). This metric is more sensitive when features are highly correlated.

6. The next step involves computing the correlation between the empirical similarity matrices and the model matrices. The within-domain (perhaps better referred to as "within-task" for consistency) model matrices appear to be based on Euclidean distances between conditions within each task. This suggests that two separate distance matrices are computed - one for each task. I recommend consolidating all distances into a single matrix, where the first set of rows (e.g., 1–29, assuming 30 unique conditions per task) corresponds to the first task, and the subsequent rows (e.g., 30–60) correspond to the second task. Binning and averaging across some unique conditions could also enhance the sensitivity of the analysis.

7. This suggestion is a continuation of the above recommendation of using one distance matrix. I would recommend using a regression instead of just correlating the matrices with neural activity. This will allow task variables to compete for variance and thus allow to compare the relative strength of the representations of each task variable. I found the interpretation of the current RSA matrices with respect to neural activity quite confusing (What does it mean that neural activity was negatively

correlated to an interaction between the decision boundary and the future space?). For example,

Observed mahal matrix = $b_0 + b_1 \cdot \text{task (seed 1 vs seed2)} + b_2 \cdot \text{features (conceptual and spatial)} + b_3 \cdot \text{decision (grow vs die)} + \text{motor response (left vs right)}$

A shared features model could also be included to assess whether the feature space is truly shared. I assume that the "features" model described above exhibits strong off-diagonal across-task values, capturing both orthogonal task representations and competing for variance with the task model. However, it is important to acknowledge that this model could become significant even in the absence of a shared structure if motor preparation signals are similar across tasks.

This regression analysis should be conducted at every time step from seed presentation to feedback, with coefficient values plotted as a function of time. This would provide a clear temporal overview of when different representations emerge and whether they become correlated. For example, I would expect motor and decision variables to correlate towards the end of training but not at the beginning.

8. The reported trial rejection rate of 4% seems unusually low. What criteria were used for trial exclusion, and why is this rate so low compared to typical artefact rejection thresholds? Additionally, given that MNE is being used, why were the built-in automatic trial rejection and repair methods not utilised?

9. How did the authors account for potential eye-movement confounds? Was any control analysis implemented to ensure that task-related signals were not driven by systematic eye movements? For example, can any of the task variables be decoded from horizontal or vertical eye movements?

10. I am wondering whether participants could truly represent the geometry of the decision boundary as defined in this task or whether they relied on more heuristic strategies (e.g., categorising based on four quadrants). Notably, even participants in the "different" group achieved an average accuracy of 75%, which may indicate that they did not acquire a full representation of the decision boundary. Have the authors considered or tested alternative heuristic strategies that participants might have used based on behavioural data? I believe that the manuscript would benefit from a more comprehensive analysis of the error trials.

Minor questions and concerns:

11. When referring to the analysed trial period, the authors use the term "stimulus-locked period," but it is unclear which specific timeframe this refers to (I assume it corresponds to the period when task features are presented). I recommend using a more intuitive term, such as "task feature-locked," to improve clarity. Additionally, visually denoting the analysed window in Figure 1D would help readers better understand the timing of the analysis.

12. The introduction frames the study as investigating behavioural and neural facilitatory effects when two tasks share or differ in their underlying structure. However, in the different-task group, the decision boundary's geometry does not fundamentally change - it is merely rotated. Given this, is it accurate to describe the DifferentSt group as having fundamentally different task structures? Clarifying this distinction would strengthen the study's theoretical framing and ensure that the terminology accurately reflects the experimental design.

13. The manuscript lacks a detailed description of the generalisation component of the testing phase - it is unclear what exactly is being generalised. Did it involve transferring the decision boundary to new configurations or to a new seed but same configurations? Why was neural activity not analysed during generalisation?

14. RSA alone is not a robust test for assessing shared decision boundaries. To strengthen the analysis, I recommend complementing it with cross-generalised decoding, as demonstrated in Bernardi et al. (2020). Specifically, training a classifier on the Die vs. Grow decision boundary in the spatial task and testing it on the conceptual task (and vice versa) would provide a more direct measure of shared representations across domains.

15. What is the rationale for including separate training and testing phases in the experiment?

(Remarks on code availability)

Reviewer #3

(Remarks to the Author)

The current study considered a novel task where subjects learn and test with two different stimulus-response task domains which either share the same structure or different structure in the task. This task was examined through conducting behavior experiments, brain scanning experiments using MEG, and simulation experiments using a neural network model. The results from the behavioral experiments showed that performance with the different structure task over-performs the same structure task in generalization with novel stimulus inputs. The results from the MEG experiments revealed that (i) early brain activity reflected sensory encoding, shifting to task-based encoding through learning; (ii) the different structure group develops task-relevant representations faster than the same structure group does; (iii) neural activity in both groups shift from

stimulus-based to task-based one over time. Finally, the model simulation using a neural network was performed by considering 3 different mapping structures between the input space to the hidden state space. When the mapping is made as orthogonal for each dimension (orthogonal LVs), the learning speed becomes identical between the same structure and different structure task. When the mapping was made as similar (similar LVs), the learning speed of the same structure became slightly slower than the one for the different structure. Finally, when the mapping was made identical for each dimension (same LVs), the learning speed of the different structure became significantly slower. From the aforementioned results based on human behavioral experiments, MEG brain imaging experiments, and model simulation experiments using a neural network, the paper concludes that (i) similarity in tasks increases learning interference unless the brain can efficiently separate representations. (ii) The brain must "orthogonalize" task representations over time, which takes longer when tasks share structure.

This paper is interesting as it shows unexpected experimental results that during training, tasks sharing a common low-dimensional structure exhibited interference, leading to worse learning performance. However, this interference decreased with practice, with performance differences disappearing during the testing sessions. The reported results could draw a large attention in various fields including neuroscience, psychology, and machine learning. The methodology seems mostly valid to support the claims. Although the paper has good potential for publication in the current journal, there are some major drawbacks which should be properly addressed. I suggest the authors to revise the current manuscript majorly by following the comments shown below.

*** Major comments

(1) In the subsection 2.1, it is written that: "This interference seemed to be reduced with practice, as both groups reached equivalent classification and generalization performance during the test phase." However, it does not seem correct since Fig. 2 B shows that Diff structure case achieves better generalization performance than Same structure case. Please clarify this point.

(2) Although the paper shows interesting behavioral and brain imaging experiment evidence on possible representation learning such as the same/orthogonal hidden state representation and stimulus-bounded/task-based representation, it does not show clear neural mechanisms accounting for those findings. In order to improve this part, the authors may perform analysis on source localization of MEG images to identify which brain regions contribute to development of representations, stimulus-bounded/task-relevant and same/orthogonal hidden state representation. Also, it is very informative to investigate what sorts of interactions among specific brain regions can lead to develop those representations.

(3) In the subsection 2.2.4 it is written that: "Early after stimulus onset, neural activity in the Spatial task was more similar for configurations that were close in the task-relevant space. This was significant for both the SameSt group (180-1080 ms, 200-1060 ms, 380-710 ms and 480-1020 ms) and the DiffSt group (800-1360 ms and 150-780 ms). Although this neural activity observation is interesting, I wonder how it would be possible to model the phenomena using artificial neural network models. A problem is that the neural network model used in the current study is just a three-layer Perceptron-type network which cannot regenerate the phenomena. The authors may write how this problem can be resolved.

(4) This paper shows that cross-domain representations shift from correlated to decorrelated in the same structure condition by interpreting the results obtained through MEG experiments. The paper seems to suggest also that the same shift can be observed during neural network model learning. If this is true, this should be stated more clearly. Currently, the underlying mechanism for the shift is not well explained. The authors should provide rational explanations for this by adding further analysis of the process of the representation learning in the proposed neural network model.

(5) Although the paper states that representation learning for generalization is essential for transferring learning repeatedly, the study does not examine whether the learned representation can transfer to new tasks. It is better to include some transfer tasks in which once trained neural network models as well as human subjects adapt to new tasks with new feature mapping.

** Minor comments

(1) Please explain how the feature dimensions in the Task-Relevant Rep. can be reduced into 1-dimension.

(2) In Fig.3B, the plot of spatial within domain and that of spatial across domain are the same. What do these same plots represent? Also in the same figure, what do red, orange and purple lines represent?

(3) Differences between Empirical Matrix and Model Matrix should be clarified. Also, please explain how configurations can be organized in these matrix.

(4) In equation (1), please explain how the parameters 2.4 and -0.71 are derived.

(Remarks on code availability)

Version 1:

Reviewer comments:

Reviewer #1

(Remarks to the Author)

No further comments. I thank the authors for their diligence in addressing my comments.

(Remarks on code availability)

Reviewer #3

(Remarks to the Author)

Thank you for your thoughtful responses to my comments and related revisions in the revised manuscript.

I have minor comments regarding to your responses as shown below.

(i) Related to the responses to my major comment (1), I suggest the following.

=> In order to avoid unnecessary ambiguity, the authors are better to state explicitly in the main text that performance on new stimuli was similar across groups.

(ii) Related to the responses to my major comment (2), I suggest the following.

=> Given the absence of source localization, still the authors could clarify how the inference of “frontal” versus “sensory” sensor involvement was conjectured. Was this based on standard sensor layouts or any statistical clustering across sensor groups? Some indication (e.g., topography plots or sensor group analyses) might help readers interpret these functional shifts with appropriate caution.

(Remarks on code availability)

Reviewer #1 (Remarks to the Author):

The authors present a manuscript aimed to characterize the similarity of task encoding with neural representations throughout learning and generalization. The authors design a simple experiment aimed to manipulate the degree of interference in the task encoding/presentation, where one variant of the task has two features that are correlated (conceptual and spatial features), and the other task encodes these features independently. They trained/evaluated two separate cohorts of human participants on each of these variants, evaluating their training and testing performances. Overall, they found that humans presented with the non-interfering task variant (i.e., independent features) tended to perform better than those trained on the interfering task variant. Subsequent neural analyses (via RSA on MEG data) revealed differences across the groups. Experiments with computational models revealed an overall similar pattern, whereby artificial nets trained with correlated task features tended to learn more slowly than those trained with independent task features.

Overall, I encountered several major issues with the paper. 1) Novelty: Much the conclusions and findings from this paper are consistent with those from the human cognitive control literature, namely studies using interference tasks such as the Stroop task (Cohen et al., 1990; Freund et al., 2021). 2) Clarity of the findings (particularly the RSA results): I found that many of the findings were not as clear nor obvious as to how they were presented in the Abstract and the Introduction. 3) Utility of the model: It seems to me the model was not necessary, and that this task can be performed by a linear classifier. Below, I explain these issues in detail.

We thank the reviewer for the critical and constructive assessment of our work. We appreciate the thoughtful feedback, which has helped us identify important areas for clarification and improvement.

Major

1. Much of these results do not seem particularly novel over and above the prior literature. It is known that incongruent conditions in task interference tasks (e.g., incongruent v. congruent conditions within the Stroop task) are harder to learn, but performance differences can be attenuated with practice (Davidson et al., 2003). Moreover, other studies have also evaluated the effects of neural representations across practice/training in humans (fMRI/EEG/MEG) and models (Cole et al., 2011; Dekker et al., 2022; Flesch et al., 2023; Freund et al., 2021; Ito et al., 2022; Kikumoto et al., 2024). I recognize that while the authors referenced some of these papers, exactly what insights were uniquely afforded from their paper relative to these was not entirely clear.

The reviewer draws a connection between our work and prior studies on task interference and neural representations in cognitive control. We believe that while there are shared themes, our study addresses a distinct form of interference. Specifically, studies such as those on the Stroop task involve incongruencies in stimuli features (immediately available to perception), our study differs in that the presumed interference emerges from similarities at the level of the underlying relational structure, which must be learned or extracted via experience. This distinction is critical, as it highlights a different mechanism of interference - one that arises dynamically through the integration of task-relevant features during learning rather than from predefined task demands. Nevertheless, we agree with the Reviewer that the above mentioned literature might be relevant for the interpretation of our results and for a fair positioning of our findings in the existing literature, therefore we have included a paragraph at the end of the Introduction and Discussion, to explicit discuss how our study provides novel insights into the learning-dependent emergence of interference.

[Added paragraph]

Contrary to our expectations, the SameSt group performed worse than the DiffSt one during the training phase, indicating that the shared structure between tasks led to interference rather than facilitation. MEG analyses revealed that the representations of the two tasks were initially correlated in the SameSt group but anticorrelated in the DiffSt group. With practice, the gap in performance decreased, and the neural representations of the two tasks in the SameSt group shifted toward an anticorrelated pattern similar to that of the DiffSt group. This dynamic aligns with findings from the cognitive control literature, suggesting mechanisms by which the brain adapts to interference through representational reorganization (Musslick 2021, Garner 2023, Badre 2024). In the Discussion, we will further compare these findings with studies on cognitive control, highlighting both similarities and key differences in how interference emerges and is resolved across different paradigms.

Additionally, we have expanded the Discussion to further contrast our findings with prior work and carve out the unique contributions of our study.

[Added paragraph]

However, unlike classic cognitive control and interference paradigms such as the Stroop task (Cohen1990, Freund2021), where incongruencies between task features are inherent to the perceptual features and task demands, in our study, the interference emerged from learning the underlying structure that links features to outcomes. This distinction is critical, as it highlights a different mechanism of interference—one that arises dynamically through the similarity in integration of, otherwise unrelated, task-relevant features during learning rather than from predefined task demands. Consistent with prior work showing that practice can mitigate interference, we observed a gradual shift in neural representations: the correlated activity in the SameSt group transitioned toward an anticorrelated pattern similar to that of the DiffSt group. This adaptation aligns with theories suggesting that the brain resolves interference by progressively separating task representations through mechanisms such as pattern separation and encoding them in distinct neural subspaces (Davidson2023, Mill2023, Libby2021, Losey2024, Flesch2023, Bhandari2024, Guise2017, Weber2023, Kikumoto2024).

2. I was not entirely clear on the main findings from the RSA analysis in MEG data. The abstract says “Crucially, practice reduced this performance gap and shifted the neural representations of the tasks in the same-structure group towards anticorrelation, like those in the different-structure group.” Unless I missed it, I did not find an analysis that supported this claim. More broadly, it was hard to compare the RSA time course traces, given that the cluster time courses were sourced from the different MEG source areas. Moreover, no effort was seemed to try and interpret/understand the sources from the MEG data.

We thank the reviewer for this important point. It is not entirely clear to us which specific analysis the reviewer is referring to when noting the absence of support for the claim. If the reviewer is referring to the statement that “*practice reduced the performance gap*,” we point to the behavioral data showing a significant performance difference between the two groups during training, which disappears during the testing phase (see Figure 2). This suggests that training helped mitigate the initial advantage in the SameSt group, supporting the claim that practice reduced the performance gap.

If the reviewer is referring to the “*shift in neural representations toward anticorrelation*,” this is addressed in our RSA results. Specifically, we show that during training, the neural representations of the two tasks differ between the SameSt and DiffSt groups during training (see Figure 4 and 13). Crucially, in the Same structure group, this representation shift from a positive correlation toward an anticorrelated pattern between training and test—similar to the stable anticorrelation observed in the DiffSt group throughout training and testing. If the reviewer is referring to the similarity between the two anticorrelated representation, we now address the reviewer’s point more directly. We have now included an additional cross-validation analysis comparing the anti-correlated representation during testing phase between the two groups.

To test this, we conducted a new analysis in which we identified the significant cluster (channels and time points) in the SameSt group and tested it in the DiffSt group, and vice versa. The results show that the significant channels and time points identified in the SameSt group and tested in the DiffSt group exhibited the same negative trend ($t(26) = -4.57, p < 0.001$). Similarly, when testing the three significant clusters found in the DiffSt group within the SameSt group, we found that the fronto-central and fronto-left lateral clusters showed the same significant pattern ($t(26) = -2.77, p = 0.01$; $t(26) = -2.82, p = 0.008$), whereas the fronto-right lateral cluster was not significant ($t(26) = -0.56, p = 0.579$).

This analysis should help comparing the RSA time course traces as suggested by the reviewer. Furthermore, we now changed the term in the abstract from “like” to “resembling”.

See the figure below, paragraph 2.2.3 main text and in the supplementary materials for further details.

Cross-Group Analysis Cross-Domain Space

We have now expanded the Discussion to include speculation about the sensor locations, particularly their difference between sensory and frontal areas, which are often implicated in cognitive control processes.

If none of this is what the reviewer was looking for, we are happy to provide more evidence and perform more analyses.

3. The narrative surrounding the behavioral results seem a bit strong, given that the only difference found in the training performance between the two groups, and not during test time. Was there any difference in the RT across the two conditions?

We appreciate the reviewer's comment and now include below reaction time analyses for both the training and test phases. Although the results are not significant, they follow the same pattern as the accuracy effects. This is not surprising, as accurate responses are typically associated with faster reaction times. We now toned down parts of the manuscript regarding the behavioral interpretations. Furthermore, we now provided more behavioral analyses in response to another reviewer, please see the updated text and supplementary materials. In brief, these new analyses show that participants accuracy was sensitive to the task structure, with errors occurring more frequently near the decision boundary, which would further strengthen, in our opinion, our interpretation of our behavioral findings.

4. Utility of the model: I found the model experiments to be generally unsurprising. It seems as though the difference in the results could be explained by simply measuring the dimensionality/entropy of the input space. In other words, the greater the dimensionality of the input encoding, the easier the model would learn the task, which seems intuitive. (In essence, assuming two separate groups/models for the in two conditions (same v. diff), wouldn't the task only require a linear classifier for each group? Is an ANN truly necessary?)

We appreciate the reviewer's comment and consideration of the model utility. The reviewer is correct that the models in the different-structure condition learn more quickly in the high-dimensional input space, where the two tasks use orthogonal latent variables (Figure 6d, left), than in the lower-dimensional input space, where the two

tasks use the same latent variables (Figure 6d, right). The opposite is true for the model in the same-structure condition. We agree that these parts of the model results are quite intuitive! However, these results do not follow the pattern we see in our behavioral data, which was the motivation for exploring the less intuitive behavior of the model in the intermediate regime of “similar” latent variables across the two tasks, which can account for our behavioral findings. We now contrast more directly between the expected and unexpected behavior of the model in the text:

We began by providing the input variables to the network in perfectly orthogonal dimensions (e.g., each variable was a different “input” to the network; Figure 6C, left). This leads to identical learning speed across the networks trained with the same- and different-structure (Figure 6D-E, left). This identical learning speed is expected, because the same- and different- structures are identical to the network. Since the variables are all in orthogonal subspaces, there is no relationship between the variables in the first and second contexts -- and they do not interfere with each other.

How can we induce an appropriate bias in the network? Instead of providing the task features along orthogonal dimensions, we provide them along semi-orthogonal -- or, similar -- dimensions (Figure 6C, middle), where we quantify similarity using the cosine similarity between the vectors defining each dimension. This means, for example, that the amount of water in context 1 will be related to the x-position of the seeds in context 2. This manipulation will bias the network toward treating the different features similarly, and produce interference between the two features that must be overcome for reliable task performance. We are not aware of previous studies that have systematically manipulated the similarity between different task features in this way. Interestingly, this manipulation alone causes faster learning for tasks with different structure relative to tasks with the same structure (Figure 6D-E, middle) - and, as a result, qualitatively replicates the behavioral effect observed in our participants.

Finally, we explore the case where the network is only given two task-relevant features (Figure 6C, right). This is an extreme version of the similar feature case above, where the features are identical (i.e., perfectly aligned, or with zero angle). In this case, the same structure task is no longer functionally contextual, since the network (or subject) can ignore the contextual variable (or seed) and still perform the task perfectly. This is not the case for the different structure task, where context remains relevant. As expected, the networks trained with the same structure learn far faster than those trained with different structure tasks (Figure 6D-E, right) -- however, this is the opposite result seen in our participants, and seen for neural networks trained with only similar and not identical latent variables across the two contexts (Figure 6D-E, middle). Thus, lower-dimensional representations do not always favor faster learning for tasks with the same structure.

As to whether or not an ANN is necessary, we now clarify in the text that a linear classifier would not be able to solve either the tasks within each individual context (e.g., the top left panel of Figure 6d), or the full task -- where the network must perform interleaved trials from each context (e.g., the top two panels of Figure 6d). We now clarify this in the text:

We used nonlinear artificial neural networks because the task structure used in the experiment is not linearly separable, and therefore cannot be solved by a simpler, linear decoder.

5. Related to the above, by 50 repetitions per model equivalent to 50 random seeds? Or the number of epochs? Why are there no error bars in figure 6d?

We now clarify in the text that:

(figure 6 legend) The learning trajectories across 50 training epochs for each input structure, averaged over $n = 50$ random initializations of the model parameters. The error bars are the standard error of the mean (SEM), but are smaller than the line width.

6. I found the MEG RSA results to be a hard to make sense of. How should one interpret negative v. positive correlations with the ground truth RSA matrix? Are the results dependent on choice of distance metric (of both the RDM matrix and/or the difference between neural and task RDMs)? How do I interpret the very low effect sizes of the actual correlation traces (Fig. 4, 5)?

- We appreciate the reviewer's questions regarding the interpretation of positive and negative correlations in the RSA results. In our study, the correlation patterns can be interpreted in relation to the designed task structure: in the DiffSt group, where the two task structures were orthogonal, we observed a negative correlation between the tasks, reflecting their distinct neural representations. Conversely, in the SameSt group, the task structures were initially aligned, leading to a positive correlation, indicative of shared representations between the two tasks. We expand on these interpretations in the Discussion.
- Regarding the choice of distance metrics, we primarily used Euclidean distance for the model RDMs and correlation distance for the neural RDMs, consistent with our previous work (Menghi et al 2023 & Menghi et al 2025). While we did not systematically explore alternative distance metrics in the main analyses, we have now tested the Mahalanobis distance, in a GLM RSA analysis, in response to the second reviewer's comments—see response to reviewer 2, major point 7, below.
- Finally, regarding the effect sizes of the RSA correlations, it is important to note that the effect we are examining—abstract representations shared across different tasks—is expected to be relatively small. This aligns with prior neuroimaging RSA studies, which typically report low correlation values due to the inherent noise in neural data and the complexity of cognitive representations.

Minor

1. How do the authors interpret the location of the clusters in the MEG source, in figures 4 and 5?

We have now expanded the Discussion to include speculation about the sensor locations, particularly their difference between sensory and frontal areas, which are often implicated in cognitive control processes.

2. The discussion of learning a compressed representation that is optimal for learning the task structure seems a bit like an overgeneralization, given that the compression is from a 2D encoding of space to a 1D encoding of relative differences to the diagonal.

We argue that compression is a crucial aspect of learning in our task. Given the large number of possible stimulus-outcome associations, learning the task without identifying its underlying structure would be nearly impossible. The task requires participants to discover and rely on a lower-dimensional representation, where only the relative differences along a single task-relevant dimension matter. For example, in the conceptual task, participant can learn that when feature-1 (sun opacity) - feature-2 (water opacity) is equal to ~ 0 , the seed grows. This compression is not just an incidental feature of the task but rather a necessary strategy for successful learning.

While in our case, the compression involves reducing a 2D feature space to a 1D task-relevant encoding, similar principles apply more broadly to learning scenarios where efficient generalization depends on extracting the most relevant dimensions from high-dimensional input spaces. We now expanded the task structure paragraph to make it clearer.

3. What was the base correlation between the stimulus-bound v. task-relevant RDM matrix?

We thank the reviewer for the question. The base correlation between the stimulus-bound and task-relevant RDM matrices was: Conceptual task – 0.6 during training and 0.5 during test; Spatial task – 0.42 during training and 0.5 during test. These values have now been added to the data analysis section of the manuscript for transparency.

While these correlations indicate some degree of overlap between the stimulus-bound and task-relevant models, which is to be expected, we believe they do not pose a significant concern for interpretability. If the models were capturing essentially the same effect, we would expect the RSA results—i.e., the correlations between the neural dissimilarity matrices and each model—to show highly similar patterns. However, our results demonstrate distinct neural correlates for each model, with different temporal and spatial profiles across training and testing phases. This suggests that, despite some correlation at the model level, the brain distinguishes between these aspects of the task, allowing us to meaningfully interpret their contributions to representational dynamics.

We now explicitly address this point in the updated (methods and discussion) manuscript to clarify why the moderate correlation between the models does not undermine our main conclusions.

References

- Cohen, J.D., Dunbar, K., McClelland, J.L., 1990. On the control of automatic processes: a parallel distributed processing account of the Stroop effect. *Psychol. Rev.* 97, 332–61. <https://doi.org/10.1037/0033-295X.97.3.332>
- Cole, M.W., Etzel, J.A., Zacks, J.M., Schneider, W., Braver, T.S., 2011. Rapid Transfer of Abstract Rules to Novel Contexts in Human Lateral Prefrontal Cortex. *Front. Hum. Neurosci.* 5, 142. <https://doi.org/10.3389/fnhum.2011.00142>
- Davidson, D.J., Zacks, R.T., Williams, C.C., 2003. Stroop Interference, Practice, and Aging. *Neuropsychol. Dev. Cogn. B Aging Neuropsychol. Cogn.* 10, 85–98. <https://doi.org/10.1076/anec.10.2.85.14463>
- Dekker, R.B., Otto, F., Summerfield, C., 2022. Curriculum learning for human compositional generalization. *Proc. Natl. Acad. Sci.* 119, e2205582119. <https://doi.org/10.1073/pnas.2205582119>
- Flesch, T., Nagy, D.G., Saxe, A., Summerfield, C., 2023. Modelling continual learning in humans with Hebbian context gating and exponentially decaying task signals. *PLOS Comput. Biol.* 19, e1010808. <https://doi.org/10.1371/journal.pcbi.1010808>
- Freund, M.C., Bugg, J.M., Braver, T.S., 2021. A Representational Similarity Analysis of Cognitive Control during Color-Word Stroop. *J. Neurosci.* 41, 7388–7402. <https://doi.org/10.1523/JNEUROSCI.2956-20.2021>
- Ito, T., Klinger, T., Schultz, D.H., Murray, J.D., Cole, M., Rigotti, M., 2022. Compositional generalization through abstract representations in human and artificial neural networks. *Adv. Neural Inf. Process. Syst.*
- Kikumoto, A., Shibata, K., Nishio, T., Badre, D., 2024. Practice Reshapes the Geometry and Dynamics of Task-tailored Representations. *bioRxiv* 2024.09.12.612718. <https://doi.org/10.1101/2024.09.12.612718>

Reviewer #2 (Remarks to the Author):

Summary

This study investigates whether a shared latent structure across two tasks facilitates learning compared to tasks with distinct latent structures. The authors used MEG to measure neural representations in 54 human participants assigned to two experimental groups: one with a shared task representation and the other with a different representation. Behavioural analyses revealed an unexpected result - contrary to the authors' predictions, participants exposed to two different latent structures learned more slowly than those with distinct task structures. Representational similarity analysis (RSA) further indicated that, irrespective of whether the decision boundary was shared, participants in both groups constructed negatively correlated representations of task feature spaces, as well as the decision boundaries (grow vs. die) across tasks.

This study presents a valuable opportunity to uncover key computational mechanisms underlying the ability to utilise abstract representations in humans. It is particularly insightful in examining how partial overlap in representation - specifically in the decision boundary but not in the stimulus space - affects learning. A notable strength of the study is also the inclusion of two experimental groups, enabling direct comparisons of learning effects.

However, the manuscript lacks critical details regarding the analyses and the rationale behind key methodological decisions, which significantly hinders a thorough assessment of the reported findings. Additionally, the operationalisation of task structure as a means to test the facilitatory effects of shared representation spaces appears to be an overextension. The results suggest that the study is less about the advantages of a shared feature space and more about the challenges posed by shared decision boundaries.

While my concerns are substantial, I believe this study has the potential to make a meaningful contribution. I recommend the manuscript for publication, provided that the authors adequately address these issues.

We thank the reviewer for the positive and encouraging feedback and for the critical and constructive assessment of our work. We appreciate the thoughtful feedback, which has helped us identify important areas for clarification and improvement of the manuscript.

Major concerns:

The analysis lacks crucial methodological details and a clear rationale for key decisions. For instance, was neural activity averaged across trials, or were representational matrices computed from single-trial data? What was the number of trials per unique condition, and how many unique conditions were included? Additionally, not all the RSA matrices are being shown in detail and what is being presented lacks important details for interpretation (e.g., axis labels). It is also unclear why neural activity was only analysed during the period when spatial or conceptual features were presented, and the seed-locked period was not explored. These omissions hinder the transparency and interpretability of the findings. I have outlined further specific concerns alongside the corresponding issues below.

We appreciate the reviewer's suggestions. To clarify, we averaged neural activity across trials, which allowed us to focus on the relevant dimension for each task while averaging out the features of the irrelevant task. This approach is crucial for the analysis, as it helps isolate the key features of interest for each condition. For example, in the spatial conditions, during the testing phase, we had 28 different spatial configurations, each paired with 8 conceptual configurations (4 associated with "grow" and 4 with "die"). By averaging across trials, we ensured that the representation of each condition was robust, while reducing noise from irrelevant features.

We updated the methods and analyses sections to reflect this more clearly. Additionally, we ensured that the relevant details, such as the number of trials per unique condition, the total number of unique conditions, and a more detailed presentation of the RSA matrices (See Fig.3), are explicitly stated in the manuscript. Regarding the analysis period, we focused on the time windows when task features were presented, as these were the periods most relevant for task performance. Based on our current analysis and the results shown in Figures 4 and 5, we do not expect the seed to be parametrically different between stimuli, as the seed is presented before the stimulus (task-feature) onset (at time 0). At this point, no feature information is yet available.

1. It remains unclear how participants in the SameSt group could have constructed a shared feature representation between the two tasks. While the geometry and orientation of the decision boundary were preserved across tasks, the feature spaces should have remained orthogonal to prevent irrelevant features from influencing the processing of relevant ones (i.e., using the space feature map to solve a conceptual task). Consequently, the task design in this group does not inherently promote shared task structure but rather a shared decision readout. Referring to this condition as "same structure" in the introduction appears misleading. A task could more justifiably be described as having a "shared task structure" if the feature spaces were correlated at the single-trial level, as this could yield an explicit behavioural advantage (e.g., allowing participants to ignore one of the feature spaces). My comment is based on two key assumptions: (1) that both the spatial map and conceptual pictograms were presented simultaneously, and (2) that the values of the task-irrelevant feature space were randomly assigned on each trial. There is a mention in the Methods section that the feature spaces "were not related," and Figure 1 suggests that they were presented simultaneously. However, these details should be explicitly stated in the text to ensure clarity.

We thank the reviewer for the comment. To clarify, our definition of "task structure" follows the framework established by Braun et al. (2010), where structure is defined as a set of parameters that organize one or more tasks, creating a compressed space with lower dimensionality than the original sensory input while preserving a similar amount of information. In our study, the task structure transformation can be either identical or orthogonal, depending on the relationship between the feature spaces. We expected this structure to be shared over trials not within one trial.

In the SameSt group, the spatial configuration was presented with 8 different conceptual configurations, each linked to "grow" and "die", this was design to ensure minimal interaction between the two features spaces. See the updated methods section explaining it more clearly.

[Added Paragraph]

During the training phase, participants were exposed to 18 different configurations in both spatial and conceptual tasks. Each configuration was repeated 8 times, with a different irrelevant configuration. For example, in the spatial condition, each spatial configuration was paired with 8 conceptual configurations (4 associated with "grow" and 4 with "die"). The total number of trials in the training phase was thus 288 (18 configurations × 8 repetitions × 2 tasks).

2. Figure 3A suggests that task-relevant representations are measured as an interaction between the binary decision boundary (Grow vs. Die) and the feature space (spatial or conceptual). What is the rationale behind this approach? Why is it relevant to represent a "Die" differently if it was based on the opposite side of the feature map? Why not decode the binary decision boundary alone, given that this is the element shared in the SameSt group? As noted earlier, it seems unlikely that participants would develop a shared representation of the feature space, as doing so would likely impair performance. Perhaps there is a justification for using this interaction model, but if so, it needs to be explicitly stated.

We thank the reviewer for the comment. The task relevant representation was designed like that for two reasons:

1. **Motor Confounding:** As noted by the reviewer, both the Stimulus-Bound and Task-Relevant representations capture distances along the relevant feature dimensions, which are not directly tied to motor output. For example, in the spatial task, configurations located in the top-left and bottom-right corners are far apart in representational space, even though they elicit the same binary response (e.g., both "Die"). If neural activity were primarily driven by motor preparation, one would expect such configurations to cluster together rather than be widely separated. Additional evidence supporting this interpretation is provided in our response to the point below.
2. **Spatial Task Representation:** We also expect that participants in the spatial task would retain information regarding their relative position within the spatial feature space.

We agree with the Reviewer that these points require to be carefully and precisely conveyed in the main manuscript, which has been modified accordingly in the analysis section (Section 4.6.1).

[Added paragraph]

"Second, the Task-relevant Representation involves computing distances along a single dimension computed based on task-relevance, by subtracting feature 2 from feature 1 , offering insights into the core aspects of the task structure. By measuring distances in this way, we can ensure that the representation reflects task-specific features rather than motor or spatial biases"

3. As currently described, the text suggests a potential confound between the task-relevant representation shared across domains and motor preparation. As mentioned earlier, the shared (across-domain) task-relevant representation appears to capture a combination of the decision boundary (Die vs. Grow) and the feature space used across tasks. The Die vs. Grow boundary directly corresponds to the two response buttons (e.g., left vs. right). Was the button-to-decision mapping randomised throughout the experiment? This issue needs to be explicitly acknowledged, and the procedures used to mitigate it should be detailed. If randomisation was not implemented, all "across-domain" analyses based on the feature-locked period are likely capturing motor preparation signals rather than genuine task-related representations.

We agree with the reviewer that it is important to consider potential motor confounds in interpreting the task-relevant representation. We note that the response mapping was fixed across trials, however, we believe it is unlikely that our results can be attributed to motor preparation for several reasons.

First, the task-relevant representation reflects distances along the relevant feature dimension, which does not correspond it's motor output. For instance, in the spatial task, configurations located in the top-left and bottom-

right corners are far apart in representational space, yet they share the same binary response (e.g., both "Die"). If the representation were driven by motor preparation alone, one would expect configurations that share the same motor output to cluster together, rather than being widely separated.

Second, it is unclear why motor responses would be parametrically modulated by the continuous distances between stimuli, as the motor response itself was binary and unrelated to the spatial or conceptual distance between configurations.

Furthermore, we observed no significant correlation between the 2-dimensional representational distances and the distances in the expected outcomes. In the spatial task, the correlation coefficients were $r = 0.09$ and $r = 0.07$ for the 'same' and 'different' structures, respectively. In the conceptual task, the correlation was $r = 0.07$ in both the same and different structures. In the testing session, these values remained weak, with $r = -0.09$ for the same structure and $r = -0.08$ for the different structure. These findings suggest that the 2-dimensional and task relevant representational distances do not align significantly with the expected outcomes across tasks and structures.

4. What is the rationale for using a searchlight procedure, and what does the procedure entail in detail? Specifically, how was the cross-validation implemented within the searchlight protocol, and how were the statistical tests corrected for multiple comparisons? Additionally, how did the results compare when the analysis was conducted simultaneously across all sensors? I recommend providing a more detailed description of these methodological choices, along with justifications for each decision.

We thank the reviewer for the suggestions. We chose to use a searchlight approach because we expected the effects of interest to be small and localized, this is especially true in the cross-task analyses. The searchlight method allows us to conduct pairwise comparisons between adjacent regions and capture these localized effects more accurately. This method also facilitates the identification of sensors where task-related representations are prominent. To ensure the robustness of our findings, we also included results from an analysis conducted across all sensors (as opposed to using the searchlight procedure) in the updated supplementary materials (and here below). Importantly, the results from this whole-sensor analysis point in the same direction as those obtained with the searchlight approach, supporting the overall consistency of our findings. We have now added a mention of this analysis in the revised analysis section of the manuscript for clarity. See figures below and supplementary materials for a longer description.

To address the concern about cross-validation, we leveraged the advantage of having two separate participant groups, which allowed us to perform a robust cross-validation between them. Specifically, as shown in Figure 15 of the supplementary materials, we cross-validated the effects observed in one group by testing them in the other group. In this approach, relevant clusters were selected from one group and then tested for their consistency in the second group.

Since the tests we performed are independent, there was no need for multiple comparison corrections within this procedure. All the single analysis are spatio-temporal cluster based corrected.

TRAINING

Stimulus-Bound Representation

This figure shows the results of the cluster permutation based on time domain only between neural dissimilarity matrices and model RDMs. For each group (Same Structure and Different Structure), we computed the correlation between brain activity patterns (across sensors) during the training session and three model RDMs: spatial, conceptual, and cross-domain computed on the stimulus-bound space. This analysis shows similar results with the searchlight.

Task-Relevant Representation

This figure shows the results of the cluster permutation based on time domain only between neural dissimilarity matrices and model RDMs. For each group (Same Structure and Different Structure), we computed the correlation between brain activity patterns (across sensors) during the training session and three model RDMs: spatial, conceptual, and cross-domain computed on the task-relevant space. This analysis shows results in the same direction with the searchlight.

TESTING

Stimulus-Bound Representation

This figure shows the results of the cluster permutation based on time domain only between neural dissimilarity matrices and model RDMs. For each group (Same Structure and Different Structure), we computed the correlation between brain activity patterns (across sensors) during the testing session and three model RDMs: spatial, conceptual, and cross-domain computed on the stimulus-bound space. This analysis shows similar results with the searchlight.

Task-Relevant Representation

This figure shows the results of the cluster permutation based on time domain only between neural dissimilarity matrices and model RDMs. For each group (Same Structure and Different Structure), we computed the correlation between brain activity patterns (across sensors) during the testing session and three model RDMs: spatial, conceptual, and cross-domain computed on the task-relevant space. This analysis shows similar results with the searchlight.

5. The procedure for computing distance/correlation matrices remains unclear. The analysis begins with the computation of an observed (neural) similarity matrix - presumably equivalent to a correlation matrix? However, Figure 3B presents r^{-1} values, which raises questions. Why is r^{-1} shown instead of raw correlation values (I recommend plotting both correlation and distance matrices in colour with appropriate colour bars for better clarity)? Additionally, is the observed correlation matrix computed between condition averages? How many unique conditions are there? Are "configurations" considered unique conditions? Furthermore, is this process performed separately for each task (spatial and conceptual)? This needs to be explicitly stated and all RSA model matrices need to be shown. I also suggest using Mahalanobis distance instead of correlations for the observed matrices (with covariance estimated from the data). This metric is more sensitive when features are highly correlated.

We thank the reviewer for the detailed and thoughtful feedback. The analysis indeed begins with the computation of a neural dissimilarity matrix, which we now describe more clearly in the revised text. Specifically, we use correlation distance ($1 - \text{Pearson's } r$) to compute dissimilarities between neural response patterns, consistent with our prior work using similar task structures (Menghi et al., 2023; Menghi et al., 2024). While Mahalanobis distance can be more sensitive in cases of high feature correlation, we opted for correlation distance to maintain consistency with these previous studies. We used Mahalanobis distance in two GLM RSA example analysis as suggested (see the response to the reviewer's point below).

The matrices presented in Figure 3B are meant to illustrate how distances are calculated across configurations for within- and cross-domain tasks. These are schematic examples designed for clarity and do not reflect the full high-dimensional matrices computed at each time point and sensor in the actual analysis. We have now clarified this point in the main text and figure caption.

Regarding the observed neural dissimilarity matrices: these are computed separately for each task (spatial and conceptual) and based on averages across trials for each unique configuration, which allows us to reduce noise and average out irrelevant feature variation. This averaging step is crucial as it allows us to isolate distances along the relevant task dimension while minimizing the influence of irrelevant dimensions. We now make this procedure more clear.

We have now explicitly stated the number of unique configurations in both the methods and analysis sections: 18 configurations were used during training and 28 during test. Each configuration was repeated multiple times (8 times in training, 8 times in test), each configuration was paired with 8 different irrelevant configurations (4 mapped to "grow", 4 to "die"). For example, one spatial configuration was paired with 8 different conceptual configurations 4 associated with grow and 4 with die outcomes. These pairings were counterbalanced to prevent systematic associations.

6. The next step involves computing the correlation between the empirical similarity matrices and the model matrices. The within-domain (perhaps better referred to as "within-task" for consistency) model matrices appear to be based on Euclidean distances between conditions within each task. This suggests that two separate distance matrices are computed - one for each task. I recommend consolidating all distances into a single matrix, where the first set of rows (e.g., 1–29, assuming 30 unique conditions per task) corresponds to the first task, and the subsequent rows (e.g., 30–60) correspond to the second task. Binning and averaging across some unique conditions could also enhance the sensitivity of the analysis.

We thank the reviewer for this suggestion. While combining all distances into a single matrix may appear to ease the analysis, doing so would introduce interpretational challenges. Specifically, merging the spatial and conceptual task matrices would necessarily include cross-task distances—i.e., distances between spatial and conceptual configurations—which are computed under a different representational assumption. Including these in a single correlation calculation would merge within-task structure with across-task relationships, making it difficult to disentangle the contributions of each.

Alternatively, excluding the cross-task distances from the combined matrix would yield a general correlation that reflects an aggregate of both task-specific representational structures, but would not allow us to isolate how well each model fits each task individually. As our goal is to assess the representational structure specific to each task (spatial and conceptual), maintaining separate RSA computations provides a clearer and more interpretable account of task-specific coding.

7. This suggestion is a continuation of the above recommendation of using one distance matrix. I would recommend using a regression instead of just correlating the matrices with neural activity. This will allow task variables to compete for variance and thus allow to compare the relative strength of the representations of each task variable. I found the interpretation of the current RSA matrices with respect to neural activity quite confusing (What does it mean that neural activity was negatively correlated to an interaction between the decision boundary and the future space?). For example,

Observed mahal matrix = $b_0 + b_1 \cdot \text{task (seed 1 vs seed2)} + b_2 \cdot \text{features (conceptual and spatial)} + b_3 \cdot \text{decision (grow vs die)} + \text{motor response (left vs right)}$

A shared features model could also be included to assess whether the feature space is truly shared. I assume that the "features" model described above exhibits strong off-diagonal across-task values, capturing both orthogonal task representations and competing for variance with the task model. However, it is important to acknowledge that this model could become significant even in the absence of a shared structure if motor preparation signals are similar across tasks.

This regression analysis should be conducted at every time step from seed presentation to feedback, with coefficient values plotted as a function of time. This would provide a clear temporal overview of when different representations emerge and whether they become correlated. For example, I would expect motor and decision variables to correlate towards the end of training but not at the beginning.

We thank the reviewer for the thoughtful and detailed suggestion. The negative correlation should be interpreted in light of the task design. In the SaSt group, the correlation is initially positive, reflecting the shared structure across tasks. However, as learning progresses, this representation gradually shifts toward an anticorrelated pattern between training and test. This pattern mirrors the stable anticorrelation observed throughout training and testing in the DiffSt group, where the task structures are orthogonal.

We agree that using a regression framework to disentangle the relative contributions of different representational components is an important goal in representational analyses. However, implementing a full GLM as proposed—especially across time points and sensors—presents several critical challenges that would both complicate the analysis considerably and reduce interpretability.

First, the model would require the construction of several competing regressors, including:

1. The 2-dimensional spatial distance between two trials (Regressor 1),
2. The 2-dimensional conceptual distance between two trials (Regressor 2),
3. The cross-domain spatial–conceptual distance between two trials (Regressor 3),
4. A task-relevance indexing when both trials are spatial (Regressor 4),
5. A task-relevance indexing when both trials are conceptual (Regressor 5)
6. A task-relevance indexing when 1 trial is spatial and 1 is conceptual for the cross-tasks (Regressor 6)
7. Interaction between regressor 1 and 4 to select the relevant spatial task distances (purple square in the figure below)
8. Interaction between regressor 2 and 5 to select the relevant conceptual task distances (yellow square in the below)
9. Interaction between regressor 3 and 6 to select the cross-task trial distances (red square in the figure below)

plus the equivalent set of regressors for the task-relevant (1 dimensional) distances. This would lead to a very conservative model with 15 regressors.

Crucially, many of these regressors would have limited applicability depending on the task. For instance, the spatial interaction regressors -7, computing distance between spatial-relevant trials (figure below, example on the left) would be zero for all trials where the distance between two conceptual trial or distance between a spatial and a conceptual task are computed, leading to 0 entries that are indistinguishable from valid 0-distance values (e.g., same location stimuli), or would need to be replaced with NaNs—potentially excluding over two-thirds of the data. This pattern would repeat for each task-specific regressor, or the cross-domain one, where values would be 0/NaN when distances between spatial trials or conceptual trials are computed. Furthermore, regressor 3 computing the distances between spatial relevant and conceptual relevant trials would be NaN everytime the distance between two spatial trials or two conceptual trials are computed (non red square right figure below).

This figure is a schematic illustration of the concatenated representational dissimilarity matrix (RDM) used to examine within- and across-task interactions. Each block represents a pairwise dissimilarity between stimulus configurations. The upper-left quadrant reflects distances between configurations in the spatial task (purple color), while the lower-right quadrant reflects distances within the conceptual task (orange color). The off-diagonal quadrants (upper-right) capture cross-task distances (red color), representing pairwise dissimilarities between spatial and conceptual configurations.

Moreover, this GLM also assume that we are working with single trials, where both task-relevant and task-irrelevant features are presented together. Given the design we have an imbalance in the number of relevant and not relevant conditions—18 or 28 configurations for the relevant task versus only 8 configurations for the irrelevant mappings (used as repetition for task relevant conditions)—would artificially inflate the apparent variance explained by the irrelevant features, making comparison across regressors misleading.

This GLM would also need to be fit at each sensor and time point, dramatically increasing computational complexity. In contrast, our RSA approach allows us to compute similarity relationships and compare them to theoretically motivated models in a straightforward and interpretable way.

Nonetheless, we agree with the reviewer that a regression framework can be valuable in some contexts, and we explored a similar approach in our analysis. Specifically, we averaged across trials belonging to the same configurations and ran the analysis separately for spatial and conceptual tasks (only for the testing session). In this process, we made both 2-dimensional and 1-dimensional distances compete by incorporating them into a GLM framework. To quantify the distances in the neural matrix, we used Mahalanobis distance across all sensors to calculate the dissimilarity matrices and then used a GLM to assess the competition between the stimulus-bound model (2-dimensional distances) and the task-relevant model (1-dimensional distances).

We observed that, unsurprisingly, the stimulus-bound model accounted for most of the variance and explained the majority of the effects. This outcome is not entirely surprising, as the stimulus-bound representation is stronger because it reflects the visual information on screen rather than an abstract manipulation. Furthermore, we do not expect the stimulus-bound and task-relevant models to fully explain the data in isolation. Rather, we hypothesize that both models likely co-exist in parallel, each contributing to the observed neural representations.

This figure represent the results from a GLM-based RSA exploratory analysis comparing stimulus-bound and task-relevant models. To assess the relative contributions of task-relevant and stimulus-bound features to neural representations, we averaged responses across trials and computed Mahalanobis distances across all MEG sensors. Separate dissimilarity matrices were constructed for the spatial and conceptual tasks during the testing session. A general linear model (GLM) was then used to assess the competing variance explained by two predictors: a stimulus-bound model (reflecting within-task physical distances between stimuli) and a task-relevant model (reflecting distances along the decision-relevant dimension). The plots show the resulting beta weights over time. As shown, the stimulus-bound model accounts for most of the variance in the neural data.

Given this result, while the GLM approach does provide insights, it does not offer a compelling advantage over our original RSA approach, especially when considering the complexity and interpretability of the models. However, we plan to further explore simplified variants of this GLM approach in future work, especially for more specific hypotheses.

In the revised manuscript, we have now clarified our choice of correlation-based RSA. This should provide greater clarity and help avoid any potential confusion regarding the interpretations of the models.

8. The reported trial rejection rate of 4% seems unusually low. What criteria were used for trial exclusion, and why is this rate so low compared to typical artefact rejection thresholds? Additionally, given that MNE is being used, why were the built-in automatic trial rejection and repair methods not utilised?

We thank the reviewer for this comment. We did not use MNE's built-in automatic artifact rejection and repair methods, instead, we employed conservative, manually defined peak-to-peak thresholds based on visual inspection of the data. As noted in the MNE documentation, "The values that are appropriate are dataset- and hardware-dependent, so some trial-and-error may be necessary to find the correct balance between data quality and loss of power due to too many dropped epochs." Rather than applying an arbitrary threshold across participants, we inspected each dataset individually to ensure a consistent and tailored approach. The resulting trial rejection rate averaged 4%, corresponding to approximately 30 discarded trials per participant.

9. How did the authors account for potential eye-movement confounds? Was any control analysis implemented to ensure that task-related signals were not driven by systematic eye movements? For example, can any of the task variables be decoded from horizontal or vertical eye movements?

During the visual artifact rejection process, we discarded trials in which eye movements were recorded, as these could potentially contaminate the neural signals of interest. However, we did not record eye tracking data in this study, so we were unable to perform a specific analysis of eye movements or control for any systematic eye movement confounds in the way that would have been ideal. We also want to underlie that the two tasks were counterbalanced in positions on the screen across participants, suggesting that the direction of eye movement should not impact our results. We now acknowledge this limitation in the Methods section.

10. I am wondering whether participants could truly represent the geometry of the decision boundary as defined in this task or whether they relied on more heuristic strategies (e.g., categorising based on four quadrants). Notably, even participants in the "different" group achieved an average accuracy of 75%, which may indicate that they did not acquire a full representation of the decision boundary. Have the authors considered or tested alternative heuristic strategies that participants might have used based on behavioural data? I believe that the manuscript would benefit from a more comprehensive analysis of the error trials.

We appreciate the reviewer's insightful question regarding whether participants truly represented the geometry of the decision boundary or relied on heuristic strategies, such as categorizing stimuli into quadrants. To address this concern, we conducted a more detailed analysis of performance patterns.

First, we examined whether performance varied with distance from the decision boundary. Our results confirm that accuracy increased as stimuli were positioned farther from the boundary, indicating that participants were sensitive to the task structure rather than relying on a rigid heuristic strategy.

Additionally, we assessed whether this pattern was consistent across training and test phases. If participants had used a heuristic approach (e.g., a quadrant-based rule), we would expect a shift in error patterns when task demands changed. However, we found that the relationship between performance and boundary distance remained stable across both phases, further supporting the idea that participants acquired and applied a structured representation.

While it remains possible that participants employed approximate strategies rather than a fully optimal decision rule, the observed error distributions suggest that their behavior was shaped by the continuous decision boundary rather than a discrete heuristic. We now discuss this in more detail in the revised manuscript (see Supplementary Materials and Behavioral Results section).

Minor questions and concerns:

11. When referring to the analysed trial period, the authors use the term "stimulus-locked period," but it is unclear which specific timeframe this refers to (I assume it corresponds to the period when task features are presented). I recommend using a more intuitive term, such as "task feature-locked," to improve clarity. Additionally, visually denoting the analysed window in Figure 1D would help readers better understand the timing of the analysis.

We thank the reviewer for this helpful suggestion. We agree that "task feature-locked" provides a clearer description of the analysed period and have updated the manuscript accordingly to improve clarity. However, we did not update Figure 1D to reflect this terminology change, as the figure was intended primarily as a high-level schematic for the task itself.

12. The introduction frames the study as investigating behavioural and neural facilitatory effects when two tasks share or differ in their underlying structure. However, in the different-task group, the decision boundary's geometry does not fundamentally change - it is merely rotated. Given this, is it accurate to describe the DifferentSt group as having fundamentally different task structures? Clarifying this distinction would strengthen the study's theoretical framing and ensure that the terminology accurately reflects the experimental design.

We thank the reviewer for raising this important point. While it is true that the decision boundary in the Different Structure group is rotated rather than entirely redefined, this rotation results in an orthogonal relationship between the task structures, this implies that the two tasks require projections into different axes of decision-relevant information. In this sense, the task structures are indeed different, as the dimensional mapping required to perform one task cannot be linearly transformed into the other without a change in representational space.

To address the reviewer's concern, we now explicitly state that the mapping between stimuli and responses reflects orthogonal structural mappings in the methods section.

13. The manuscript lacks a detailed description of the generalisation component of the testing phase - it is unclear what exactly is being generalised. Did it involve transferring the decision boundary to new configurations or to a new seed but same configurations? Why was neural activity not analysed during generalisation?

We thank the reviewer for the helpful comment. The generalisation phase involved applying the same decision boundary to new stimulus configurations not seen during training. Thus, participants were required to generalise the learned task structure to novel inputs. Importantly, both previously seen and new configurations were analysed together in our neural analyses. Neural activity during the generalisation (i.e., testing) phase was indeed included in our RSA analyses. We have now clarified this point in the revised results section, to avoid any confusion.

14. RSA alone is not a robust test for assessing shared decision boundaries. To strengthen the analysis, I recommend complementing it with cross-generalised decoding, as demonstrated in Bernardi et al. (2020). Specifically, training a classifier on the Die vs. Grow decision boundary in the spatial task and testing it on the conceptual task (and vice versa) would provide a more direct measure of shared representations across domains.

We thank the reviewer for the insightful suggestion. We agree that cross-generalised decoding, as used in Bernardi et al. (2020), can be a powerful approach to assess shared representations across domains. However, in our case, several factors make such an analysis less feasible and potentially unreliable. First, the continuous nature of our stimuli (e.g., positions in a 2D feature space) complicates the use of standard decoding methods, which typically perform better with discrete and clearly separable categories. Second, each stimulus contains both task-relevant and task-irrelevant features, introducing ambiguity into the classification problem. Third, the number of trials per configuration is relatively limited, which would reduce the statistical power and reliability of any decoding results—especially when attempting cross-generalisation between tasks. Finally, and critically, performing decoding on Grow vs. Die decisions would introduce a decision/motor confound. For these reasons, we chose RSA, which allows us to capture representational geometries without imposing categorical labels, as a more appropriate method for our design.

15. What is the rationale for including separate training and testing phases in the experiment?

We thank the reviewer for this question. The rationale for including separate training and testing phases was twofold. First, the training phase allowed participants to learn the task structure and establish a stable structure based on feedback. Second, the testing phase enabled us to assess participants' ability to generalise this learned structure to new configurations. Importantly, no feedback was provided during the testing phase, meaning that

performance on novel stimuli could not be attributed to continued reinforcement but instead reflects true generalisation of the learned decision rule.

Reviewer #3 (Remarks to the Author):

The current study considered a novel task where subjects learn and test with two different stimulus-response task domains which either share the same structure or different structure in the task. This task was examined through conducting behavior experiments, brain scanning experiments using MEG, and simulation experiments using a neural network model. The results from the behavioral experiments showed that performance with the different structure task over-performs the same structure task in generalization with novel stimulus inputs. The results from the MEG experiments revealed that (i) early brain activity reflected sensory encoding, shifting to task-based encoding through learning; (ii) the different structure group develops task-relevant representations faster than the same structure group does; (iii) neural activity in both groups shift from stimulus-based to task-based one over time. Finally, the model simulation using a neural network was performed by considering 3 different mapping structures between the input space to the hidden state space. When the mapping is made as orthogonal for each dimension (orthogonal LVs), the learning speed becomes identical between the same structure and different structure task. When the mapping was made as similar (similar LVs), the learning speed of the same structure became slightly slower than the one for the different structure. Finally, when the mapping was made identical for each dimension (same LVs), the learning speed of the different structure became significantly slower. From the aforementioned results based on human behavioral experiments, MEG brain imaging experiments, and model simulation experiments using a neural network, the paper conclude that (i) similarity in tasks increases learning interference unless the brain can efficiently separate representations. (ii) The brain must "orthogonalize" task representations over time, which takes longer when tasks share structure.

This paper is interesting as it shows unexpected experimental results that during training, tasks sharing a common low-dimensional structure exhibited interference, leading to worse learning performance. However, this interference decreased with practice, with performance differences disappearing during the testing sessions. The reported results could draw a large attention in various field including neuroscience, psychology, and machine learning. The methodology seems mostly valid to support the claims. Although the paper has good potential for publication in the current journal, there are some major drawbacks which should be properly addressed. I suggest the authors to revise the current manuscript majorly by following the comments shown below.

We thank the reviewer for the positive feedback and for the critical and constructive assessment of our work. We appreciate the thoughtful feedback, which has helped us identify important areas for clarification and improvement of the manuscript.

*** Major comments

(1) In the subsection 2.1, it is written that: "This interference seemed to be reduced with practice, as both groups reached equivalent classification and generalization performance during the test phase." However, it does not seem correct since Fig. 2 B shows that Diff structure case achieves better generalization performance than Same structure case. Please clarify this point.

We appreciate the reviewer's comment and acknowledge the potential confusion in our wording. The key difference between groups is not in their ability to generalize but in their ability to memorize the training configurations. While both groups performed above chance when classifying new stimuli, the DiffSt group showed better accuracy for previously seen stimuli during the test phase. This suggests that the interference observed in the SameSt group primarily affected the retention of trained examples rather than the ability to generalize to novel ones. To improve clarity, we have now revised the text to ensure that the distinction between generalization and memorization is more explicit and change the title of the figure 2B and Supplementary figure 8 to "Novelty" instead of generalization.

[Added paragraph]

First, in our behavioural data analysis, we compared participant accuracies between the two distinct groups and assessed their generalization performance on a novel set of configurations that were interleaved with old ones.

(2) Although the paper shows interesting behavioral and brain imaging experiment evidence on possible representation learning such as the same/orthogonal hidden state representation and stimulus-bounded/task-based representation, it does not show clear neural mechanisms accounting for those findings. In order to improve this part, the authors may perform analysis on source localization of MEG images to identify which brain regions contribute to development of representations, stimulus-bounded/task-relevant and same/orthogonal hidden state representation. Also, it is very informative to investigate what sorts of interactions among specific brain regions can lead to develop those representations.

We appreciate the reviewer's suggestion regarding source localization and the investigation of brain regions involved in developing different representations. During data collection, we decided to prioritize temporal resolution and the number of participants over spatial precision, opting not to acquire individual anatomical images / testing participants with an anatomical image. While this decision allowed us to maximize statistical power and capture fine-grained temporal dynamics, it also made source reconstruction less reliable. We acknowledge that identifying specific brain regions and their interactions could provide further insights into the neural mechanisms underlying representation learning. However, given the limitations in anatomical data, we believe that sensor-level analyses currently provide the most robust and interpretable results. See the updated discussion for a cautious interpretations of the sensor-space.

[Added paragraph]

Moreover, we observed that the sensors involved in the representations shifted over the course of learning, initially engaging more sensory-related sensors during training and stimulus-bound representations, transitioning toward more frontal areas during the test phase and for between-task correlations. This shift aligns with the idea that learning drives neural representations from perceptual encoding toward higher-order cognitive regions, potentially reflecting increased cognitive control, task separation and multi-tasking (Ito2022,Cole2011,Duncan2000,Sakai2008,Duan2024). In our study, we only assessed sensor-level data; therefore, any interpretation regarding the underlying anatomical sources should be taken with caution.

(3) In the subsection 2.2.4 it is written that: "Early after stimulus onset, neural activity in the Spatial task was more similar for configurations that were close in the task-relevant space. This was significant for both the SameSt group (180-1080 ms, 200-1060 ms, 380-710 ms and 480-1020 ms) and the DiffSt group (800-1360 ms and 150-780 ms).

Although this neural activity observation is interesting, I wonder how it would be possible to model the phenomena using artificial neural network models. A problem is that the neural network model used in the current study is just a three-layer Perceptron-type network which cannot regenerate the phenomena. The authors may write how this problem can be resolved.

We appreciate the reviewer's comment regarding the relationship between our neural results and the artificial neural network (ANN) model. Our goal was to use the ANN to investigate and compare how different task structures influence learning dynamics, rather than directly modeling the temporal evolution of neural similarity. Indeed, the three-layer perceptron used in our study lacks temporal dynamics and does not capture the trial-by-trial evolution of neural activity. While this is beyond the scope of the current study, we acknowledge its potential for future work and now clarify this in the revised discussion.

(4) This paper shows that cross-domain representations shift from correlated to decorrelated in the same structure condition by interpreting the results obtained through MEG experiments. The paper seems to suggest also that the same shift can be observed during neural network model learning. If this is true, this should be stated more clearly. Currently, the underlying mechanism for the shift is not well explained. The authors should provide rational explanations for this by adding further analysis of the process of the representation learning in the proposed neural network model.

We appreciate the reviewer's observation. However, we did not explicitly demonstrate such a shift in our neural network model. Our analysis focused primarily on performance similarities between the model and human participants rather than directly examining representational changes over training in the neural network. We now clarify this distinction in the revised manuscript. That said, investigating how representations evolve within the

network could provide valuable insights into the mechanisms underlying this shift. We now discuss a recent relevant study following this direction in the revised discussion section.

(5) Although the paper states that representation learning for generalization is essential for transferring learning repeatedly, the study does not examine whether the learned representation can transfer to new tasks. It is better to include some transfer tasks in which once trained neural network models as well as human subjects adapt to new tasks with new feature mapping.

We thank the reviewer for the comment. Transfer was our initial hypothesis for the study; we anticipated that participants and the neural network would leverage previously learned representations to generalize to new tasks. However, what we found, in human, was interference instead of transfer. We are currently exploring hypotheses to explain this unexpected interference, but this is beyond the scope of the current paper. In the revised manuscript, we have extended our discussion to further address why this might have occurred.

**** Minor comments**

(1) Please explain how the feature dimensions in the Task-Relevant Rep. can be reduced into 1-dimension. The reduction to a one-dimensional representation is achieved through a simple subtraction operation: $\text{feature1} - \text{feature2}$. This transformation effectively captures the task-relevant dimension by computing the difference between the two features, aligning with the task structure and simplifying the representation accordingly. We have now added this in more detail in the caption of the analysis figure and in the methods section.

(2) In Fig.3B, the plot of spatial within domain and that of spatial across domain are the same. What do these same plots represent? Also in the same figure, what do red, orange and purple lines represent?

These figures serve as examples illustrating how we calculated the Euclidean distance between different configurations. We have now clarified this in the figure caption to ensure better understanding. The red, orange, and purple lines represent different pairwise distances, which we have now described in the updated caption.

(3) Differences between Empirical Matrix and Model Matrix should be clarified. Also, please explain how configurations can be organized in these matrix.

We appreciate the request for clarification. The empirical matrix represents the neural dissimilarity matrix derived from MEG data, while the model matrix reflects the predicted distances based on stimulus-bound and task-relevant distances. The configurations in these matrices are organized such that each row and column corresponds to a specific stimulus configuration, with the values indicating the distance between pairs of configurations. We have now clarified this distinction in the methods section.

(4) In equation (1), please explain how the parameters 2.4 and -0.71 are derived.

The parameters 2.4 and 0.71 have been arbitrarily chosen to create the maps. We also updated the equation now.

REVIEWERS' COMMENTS

Reviewer #1 (Remarks to the Author):

No further comments. I thank the authors for their diligence in addressing my comments.

We thank Reviewer #1 for their time and constructive feedback, which helped us improve the clarity and quality of the manuscript.

Reviewer #3 (Remarks to the Author):

Thank you for your thoughtful responses to my comments and related revisions in the revised manuscript.

We appreciate the reviewer's thoughtful feedback which contributed to improving the manuscript

I have minor comments regarding to your responses as shown below.

(i) Related to the responses to my major comment (1), I suggest the following.

=> In order to avoid unnecessary ambiguity, the authors are better to state explicitly in the main text that performance on new stimuli was similar across groups.

We have added the requested clarification in the behavioural results section:

“Performance on new stimuli was similar across groups, showing that both SameSt and DiffSt participants generalized knowledge to novel configurations.”

(ii) Related to the responses to my major comment (2), I suggest the following.

=> Given the absence of source localization, still the authors could clarify how the inference of “frontal” versus “sensory” sensor involvement was conjectured. Was this based on standard sensor layouts or any statistical clustering across sensor groups? Some indication (e.g., topography plots or sensor group analyses) might help readers interpret these functional shifts with appropriate caution.

We have now clarified in the text that the inference is based on the standard MEG sensor layout and the spatial distribution of significant clusters. We also emphasize that, in the absence of source localization, these interpretations remain speculative and should be considered with caution.

In our study, we only assessed sensor-level data; therefore, interpretations regarding the underlying anatomical sources are based on the standard MEG sensor layout and the spatial distribution of significant clusters, and should be taken with caution.